# BET inhibition disrupts transcription but retains enhancer-promoter contact

Nicholas T. Crump [1], Erica Ballabio[1], Laura Godfrey[1], Ross Thorne[1], Emmanouela Repapi [2], Jon Kerry[1], Marta Tapia[1,6,7], Peng Hua [3], Christoffer Lagerholm[4], Panagis Filippakopoulos[5], James O. J. Davies [3] & Thomas A. Milne [1]✉

Enhancers are DNA sequences that enable complex temporal and tissue-specific regulation of genes in higher eukaryotes. Although it is not entirely clear how enhancer-promoter interactions can increase gene expression, this proximity has been observed in multiple systems at multiple loci and is thought to be essential for the maintenance of gene expression. Bromodomain and Extra-Terminal domain (BET) and Mediator proteins have been shown capable of forming phase condensates and are thought to be essential for super-enhancer function. Here, we show that targeting of cells with inhibitors of BET proteins or pharmacological degradation of BET protein Bromodomain-containing protein 4 (BRD4) has a strong impact on transcription but very little impact on enhancer-promoter interactions. Dissolving phase condensates reduces BRD4 and Mediator binding at enhancers and can also strongly affect gene transcription, without disrupting enhancer-promoter interactions. These results suggest that activation of transcription and maintenance of enhancer-promoter interactions are separable events. Our findings further indicate that enhancer-promoter interactions are not dependent on high levels of BRD4 and Mediator, and are likely maintained by a complex set of factors including additional activator complexes and, at some sites, CTCF and cohesin.

[1] MRC Molecular Haematology Unit, MRC Weatherall Institute of Molecular Medicine, NIHR Oxford Biomedical Research Centre Haematology Theme, Radcliffe Department of Medicine, University of Oxford, Oxford OX3 9DS, UK. [2] MRC WIMM Centre for Computational Biology, MRC Weatherall Institute of Molecular Medicine, Radcliffe Department of Medicine, University of Oxford, Oxford OX3 9DS, UK. [3] MRC Molecular Haematology Unit, MRC Weatherall Institute of Molecular Medicine, Radcliffe Department of Medicine, University of Oxford, Oxford OX3 9DS, UK. [4] Wolfson Imaging Centre Oxford, MRC Weatherall Institute of Molecular Medicine, University of Oxford, Oxford OX3 9DS, UK. [5] Structural Genomics Consortium, Nuffield Department of Clinical Medicine, University of Oxford, Oxford OX3 7DQ, UK. [6] Present address: The Finsen Laboratory, Rigshospitalet, Faculty of Health Sciences, University of Copenhagen, Copenhagen, Denmark. [7] Present address: Biotech Research and Innovation Centre (BRIC), Faculty of Health Sciences, University of Copenhagen, Copenhagen, Denmark. ✉email: thomas.milne@imm.ox.ac.uk

In higher eukaryotes, enhancers are DNA sequences that allow for the complex regulation of genes in different tissues and at different times[1,2]. Despite the importance of enhancers, very little is known about exactly how they function, although they have been proposed to act mainly as binding platforms for the assembly of protein complexes that can promote gene activation[1–3]. A key aspect to this assembly is the binding of sequence-specific DNA binding factors such as transcription factors (TFs). Enhancers can be situated far away from the genes they regulate[1,4]. Although not always the case[5,6], at many gene loci proximity between enhancers and promoters is thought to be essential for enhancer function and gene activation[7,8]. How these enhancer–promoter interactions are initiated and maintained is not clearly understood.

Emerging work suggests that enhancers function within larger domains, the boundaries of which are defined by the combined effects of CTCF-marked boundary regions and cohesin looping, through a process known as loop extrusion[9,10]. It is not entirely clear how these higher-order structures impact enhancer function, but generally speaking functional enhancer–promoter interactions are limited to genes within or at the edges of domains. The genome-wide existence of these more localized enhancer–promoter looping structures has been demonstrated by global chromosome conformation capture (3 C) techniques[11,12]. However, unless each sample is sequenced extremely deeply[13] (something that is not practical for most experiments), Hi-C is not able to delineate enhancer–promoter interactions at high resolution, so it is difficult to study these structures in detail genome-wide. The high complexity of Hi-C libraries and the high cost per sample and difficulty of analyzing such large datasets has meant that there have been multiple attempts to develop high-throughput methods to provide more information specifically about enhancer–promoter interactions[11,12,14,15], but the highest resolution studies focus on individual genes/enhancers using next-generation techniques, such as 4 C[16], UMI-4C seq[17], Next Generation Capture-C[18], Tri-C[19], and tiled Capture-C[20].

While chromatin looping mediated by cohesin and bounded by CTCF binding is the most common explanation for controlling large-scale chromatin structure, less is known about what stabilizes more localized enhancer–promoter contacts. Possible models include cohesin-stabilized enhancer–promoter interactions[21–23] and protein/RNA complexes bound to both the enhancer and promoter that interact with one another[24–26]. Binding of these complexes is likely initiated by key sequence-specific TFs. The presence of specific histone modifications at the enhancer is thought to contribute to one or all of these models, mainly by stabilizing the presence of specific protein complexes, such as cohesin binding to H3K4me1 or BRD4 (Bromodomain-containing protein 4) binding to H3K27ac[27–29]. Recent work from our lab also suggests that at H3K79me2/3-dependent enhancer elements (KEEs), the presence of H3K79me2/3 can help maintain open chromatin regions to facilitate the binding of sequence-specific transcription factors, and is required for enhancer–promoter interaction[30]. This could constitute a more general principle where histone modifications help regulate DNA accessibility and TF binding, and ultimately the formation of enhancer–promoter loops.

Super-enhancers are enhancers with increased enrichment for binding of TFs and coactivator complexes such as BRD4 and Mediator, and are also associated with high levels of transcriptional activity[31,32]. In cancer cells, important oncogenes are often associated with super-enhancers[33,34]. Recent work has shown that many enhancer-associated factors, such as Mediator (e.g., MED1) and BRD4, assemble into phase-separated activation complexes, and these interactions are proposed to be integral to their ability to activate transcription[3,31,35–39], but a direct requirement for phase condensate formation in

transcription has not been established[40]. Since these coactivator clusters, which assemble at enhancers (particularly super-enhancers), are also proposed to incorporate promoter-bound RNA polymerase into the condensate[31,35,38,39], it is possible that they act as a bridge between these distal DNA elements, and may have a role in initiating and/or maintaining enhancer–promoter interactions[37,41,42].

Taken together, these various strands of evidence suggest the following model: (a) loop extrusion mediated by cohesin generates higher-order chromatin structures bounded by CTCF; (b) TFs bound to enhancers and promoters assemble phase condensates made up of chromatin proteins such as BRD4 and coactivators such as Mediator; (c) histone modifications maintain accessibility for the binding of TFs and create additional affinities to further stabilize complexes; (d) the CTCF/cohesin-delimited structures create a smaller DNA compartment, increasing the frequency of random interactions between complexes bound at enhancers and promoters; and (e) the phase-separated condensates containing BRD4 and Mediator anchored at the enhancer and promoter act as a bridge to stabilize these enhancer–promoter interactions. Thus, the model posits that formation of phase condensates is a key requirement for at least a subset enhancer–promoter looping. Recent work testing this model indicated no loss of enhancer–promoter contact following degron-mediated loss of MED14, suggesting that Mediator is not responsible for these interactions[21]. However, this study used Hi-C (binned at 5 kb) and promoter capture Hi-C, which are relatively low resolution and low sensitivity techniques, so it is possible that perturbations in specific enhancer–promoter contacts may have been missed. It therefore remains unclear whether BRD4 or Mediator play any role in organizing chromatin structure and maintaining enhancer–promoter interactions at active genes.

In this paper, we directly test the role of BRD4 and Mediator in enhancer–promoter interactions by performing high resolution Next Generation Capture-C[18] in cells treated with BET inhibitors, a BRD4-degrading compound (AT1) and 1,6-hexanediol. This technique provides the greatest resolution and sensitivity of all the available 3 C methods for higher eukaryotic cells[43]. Data are generated using four-cutter restriction enzymes and are of sufficient sequencing depth that they can be reported at single restriction fragment resolution. In addition, the method is highly sensitive and reproducible, meaning that changes in interaction frequency can be analyzed quantitatively under different conditions at many genes simultaneously.

We find that reduction of BRD4 and Mediator binding at enhancers has a dramatic and rapid effect on gene expression, but enhancer–promoter looping structures remain stably intact. This suggests that the function of these activation complexes at enhancers does not involve stabilization of the enhancer–promoter interaction. Instead, we see evidence of CTCF and cohesin binding at many enhancers, indicating that these complexes can stabilize and maintain looping structures even in the presence of reduced transcription and activation complexes at the enhancer. Finally, our results demonstrate that stabilization of enhancer–promoter interactions and promotion of transcription are separable events, and that the presence of an enhancer–promoter loop is not sufficient for the maintenance of transcription.

## Results

### BET and Mediator proteins bind to active enhancers and promoters in leukemia cells. BRD4 and Mediator binding are key characteristics of enhancers, particularly super-enhancers, which are defined as having high levels of these proteins over extended regions[32–34]. We analyzed levels of BET-domain

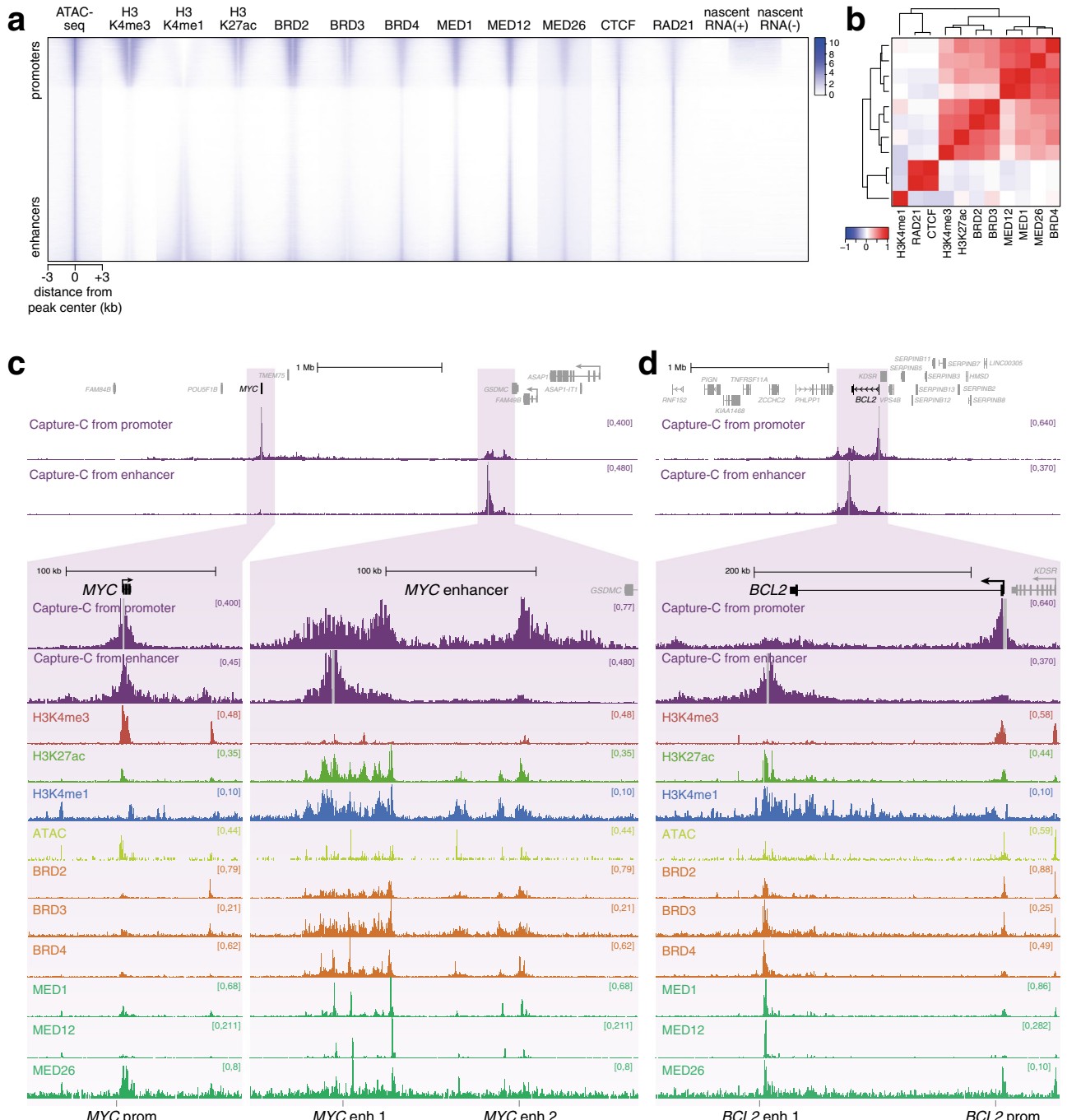

**Fig. 1 BET proteins and Mediator are a key feature of enhancers. a** Heatmap comparing levels of histone modifications, chromatin proteins and stranded nascent RNA-seq at ATAC-seq peaks in SEM cells. Peaks are ranked based on the relative levels of H3K4me3 and H3K4me1, placing promoter-like ATAC-seq peaks towards the top and enhancer-like ATAC-seq peaks towards the bottom. **b** Pearson correlation coefficients for ChIP-seq data at ATAC-seq peaks shown in **a**. Dendrogram shows hierarchical clustering of datasets. Source data are provided as a Source Data file. **c** Capture-C, ChIP-seq and ATAC-seq at the *MYC* gene and enhancer region in SEM cells. Capture-C was conducted using the *MYC* promoter or enhancer region as the viewpoint, indicated by vertical gray bars, and is displayed as the mean of three biological replicates. Locations of primers used for BRD4/Mediator ChIP-qPCR are shown at the bottom of the figure. **d** Capture-C, ChIP-seq, and ATAC-seq data at *BCL2*, as in **c**.

(Bromodomain and Extra-Terminal domain) proteins and Mediator subunits at ATAC peaks genome-wide in the leukemia cell line SEM, with peaks ranked by the relative levels of H3K4me3 and H3K4me1, thereby separating promoter (top of heatmap) and enhancer (bottom of heatmap) loci (Fig. 1a). BET-domain proteins (i.e., BRD2, BRD3, and BRD4) and Mediator subunits MED1, MED12, and MED26 all showed an enrichment at both promoter and enhancer ATAC peaks,

comparable to the distribution of H3K27ac (Fig. 1a, Supplementary Fig. 1a). Consistent with the idea that BRD4 physically interacts with Mediator[44–47], BRD4 binding positively correlated with all three Mediator subunits at ATAC peaks (Fig. 1b). In contrast, although they appear to overlap at a subset of loci (Fig. 1a), CTCF and the cohesin subunit RAD21 clustered separately from BRD4/Mediator (Fig. 1b), showing a distinct distribution at ATAC peaks, with similar levels

at promoters, enhancers and other accessible regions (Fig. 1a, Supplementary Fig. 1a).

Since BRD4 is associated with the enhancers and promoters of highly transcribed genes (Fig. 1a and Supplementary Fig. 1b, c)[32–34], we wanted to examine its role in enhancer function in more detail. Two classic BRD4-dependent genes are *MYC* and *BCL2*[48,49]. We used the high resolution 3 C technology Next Generation Capture-C[18] to identify enhancers for these genes based on their ability to interact with their promoters (Fig. 1c, d)[30]. Our Capture-C experiments in SEM cells revealed a high frequency of interaction between the *MYC* promoter and a large (~200 kb) region, composed of two major domains, located ~1.7 Mb away. This long-distance interaction has also been observed and characterized in several other cell types[22,23,50,51]. Reciprocal Capture-C from the more proximal of the two enhancer regions demonstrated contact with the *MYC* promoter, avoiding intervening regions, as well as a relatively weak interaction with the more distal enhancer domain (Fig. 1c). The enhancer is marked with broad domains of H3K27ac, BET proteins and Mediator, with much higher levels than at the *MYC* promoter (Fig. 1c, lower). In addition, we observed multiple peaks of chromatin accessibility by ATAC-seq (Fig. 1c), and enrichment for multiple transcription factors (Supplementary Fig. 1d). These characteristics are consistent with the region acting as a strong enhancer to regulate *MYC* expression; indeed, it is defined as a super-enhancer following established criteria[32–34].

We and others have previously demonstrated the presence of an enhancer at the 3' end of *BCL2* in SEM cells[30,52,53], and Capture-C from the enhancer illustrates its interaction with the *BCL2* promoter (Fig. 1d). As at *MYC*, this region is identified as a super-enhancer and is marked by elevated levels of H3K27ac, BRD4 and Mediator, relative to the *BCL2* promoter (Fig. 1d, lower).

To investigate the association of BRD4 with enhancer–promoter interactions on a larger scale, we analyzed Capture-C data for the promoters of 62 genes (Supplementary Data 1). We used ChIP-seq data for a number of enhancer-associated features to ask whether these features were commonly associated with regions showing an increased frequency of interaction with gene promoters. Indeed, H3K4me1, H3K27ac, BRD4, and MED1 were all associated with a higher frequency of promoter contact, compared to the average interaction frequency across the interaction domains (Supplementary Fig. 1e). In contrast, the repressive histone modification H3K27me3 was found at loci with reduced promoter contact frequency (Supplementary Fig. 1e). This analysis revealed that BRD4 and Mediator binding is associated with a higher frequency of interaction with promoters, potentially implicating these proteins in stabilizing enhancer–promoter contacts at these and likely other genes genome-wide.

**BET inhibition is associated with large transcriptional changes at key oncogenic gene targets**. In order to investigate the role of BRD4 in enhancer function, we used the small molecule inhibitor IBET-151 (IBET), which disrupts binding of BET protein bromodomains to acetyllysine residues. IBET is known to disrupt transcription, so we wanted to use a short treatment time to limit secondary effects from regulatory events downstream of initial transcriptional changes. We used qRT-PCR to assess how quickly IBET treatment affects gene expression. However, the stability of mature transcripts means that there is often a delay between decreased transcription and changes in mRNA levels (Supplementary Fig. 2a). We therefore used primers against intronic sequences to quantify levels of the more labile pre-mRNA. Strikingly, we observed very rapid changes in transcription, with levels of *MYC* pre-mRNA decreasing after only 15 min IBET treatment (Fig. 2a, left). Levels of *BCL2* were also

sensitive to IBET treatment, with ~50% loss after 90 min (Fig. 2a). In contrast, a comparable decrease in mature *BCL2* mRNA was not detected before 3 h (Supplementary Fig. 2a). IBET also resulted in the similarly rapid upregulation of a number of genes (Fig. 2a, right).

To assess the direct effects of BRD4 inhibition, we chose a 90 min IBET treatment time. We analyzed the global transcriptional response to IBET by sequencing nascent RNA[54], which provides a much more direct measure of transcriptional output compared to steady state RNA-seq (Fig. 2b, Supplementary Data 2). As a comparison, we also sequenced nascent RNA following 24 h IBET treatment (Fig. 2b, Supplementary Data 2). The number of differentially expressed genes was comparable after 90 min and 24 h IBET treatment, and there was reasonable correlation between the intensity of the changes under each condition (R = 0.63, Fig. 2b, right), suggesting that the shorter treatment time is sufficient to capture the immediate effects of BET-domain inhibition.

Surprisingly, given the role of BRD4 in promoting transcription, we observed similar numbers of up- and downregulated genes, several of which we confirmed by qRT-PCR (Fig. 2a, Supplementary Fig. 2b). Upregulated genes were enriched for biological pathways associated with a response to chemical stimulus (Supplementary Fig. 2c) indicating that these may be an indirect response to drug treatment, consistent with previously published work[55].

We confirmed the genome-wide reduction of BRD4 binding after 90 min incubation with IBET by reference-normalized ChIP-seq (Fig. 2c, d) and ChIP-qPCR (Supplementary Fig. 2d). At the *MYC* and *BCL2* enhancers, BET inhibition was associated with reduced transcription (Fig. 2e, Supplementary Fig. 2b) as well as a decrease in BRD4 binding (Fig. 2f, Supplementary Fig. 2d). Consistent with the reported interaction of BRD4 and Mediator[44–47], treatment with IBET also resulted in dissociation of Mediator subunits from chromatin, with reductions in MED1 and MED12 binding at the *MYC* enhancer (Supplementary Fig. 2d). Thus, targeting BRD4 reduces the level of BRD4 and Mediator binding to chromatin, and this is associated with a reduction in transcription.

As a comparison to IBET treatment, which partially reduced BRD4 binding to chromatin, we decided to make use of the PROTAC (proteolysis-targeting chimera) bifunctional molecule AT1. This consists of the BET inhibitor JQ1 conjugated to a von Hippel-Lindau (VHL) ligand, resulting in targeted degradation of BRD4[56]. BRD4 protein was not detectable by Western blotting after 24 h PROTAC treatment (Supplementary Fig. 2e) and expression of several BRD4-dependent genes, including *MYC* and *BCL2*, was downregulated (Fig. 2e, Supplementary Fig. 2 f). AT1 treatment resulted in a strong loss of BRD4 chromatin association, detectable both by reference-normalized ChIP-seq (Fig. 2d, f, g) and ChIP-qPCR (Supplementary Fig. 2 g). In addition, we observed a global reduction in MED1 chromatin association (Fig. 2f, g, Supplementary Fig. 2g, h), consistent with the effect observed with IBET (Supplementary Fig. 2d). Notably, while the loss of BRD4 binding to chromatin was more dramatic following AT1 treatment compared to IBET (Fig. 2d, f), the downregulation of transcription was comparable for the genes analyzed (Fig. 2e, Supplementary Fig. 2f), suggesting that even a moderate decrease in BRD4 binding is sufficient to perturb its transcriptional role.

**BET inhibition has very little effect on enhancer–promoter looping**. BRD4 and MED1 have recently been shown to be present in phase condensates in the nucleus[31,35,42], and this clustering is proposed to be important for the function of super-enhancers, potentially by mediating interactions with target gene promoters. In

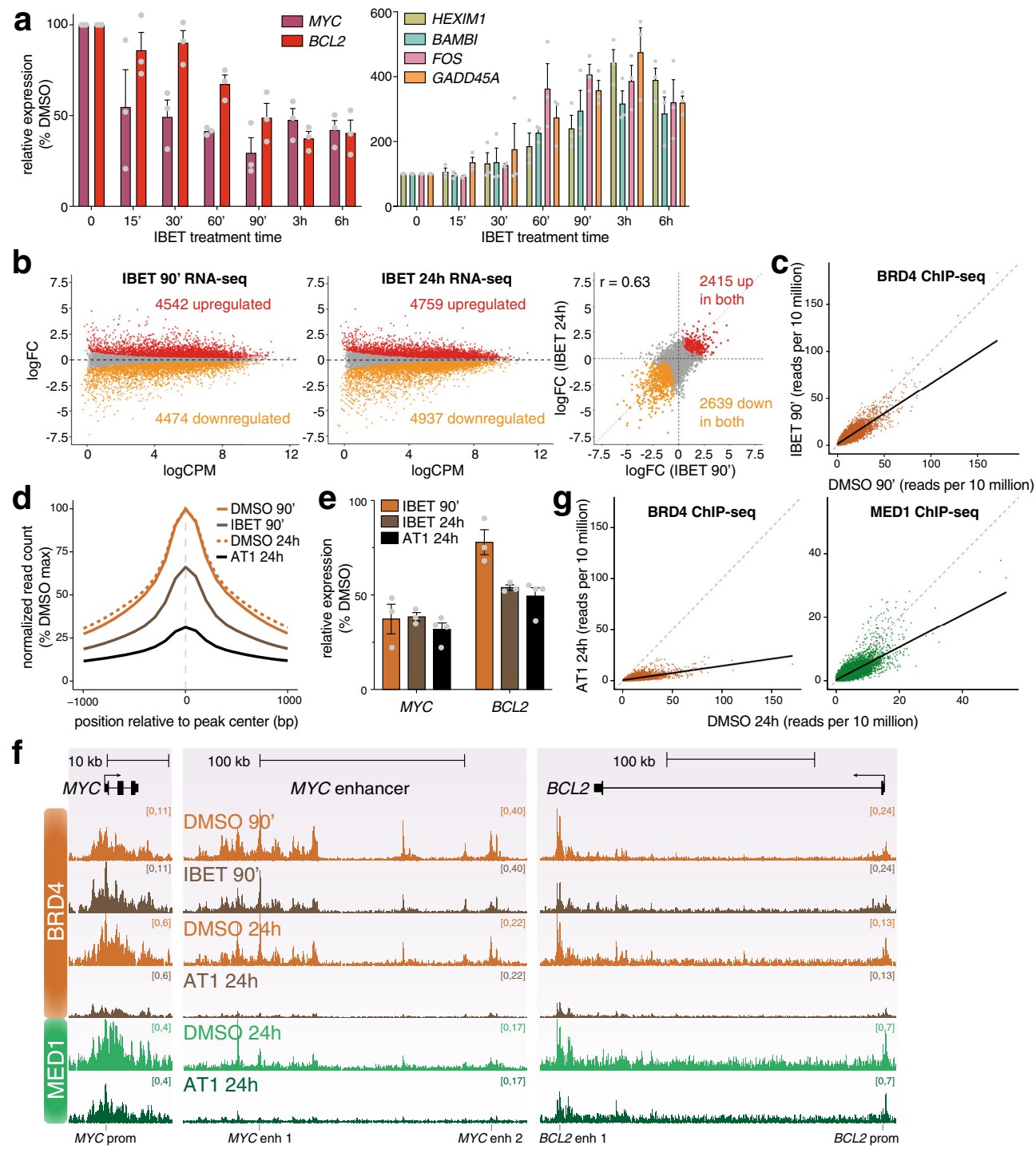

previously published work using high resolution imaging, treatment of cells with the small molecule inhibitor JQ1, which, like IBET-151, disrupts binding of the BRD4 BET domain to acetyllysine residues, prevented clustering of Mediator[35], indicating that association with chromatin is integral to phase condensation of these proteins. We therefore asked whether inhibition of BRD4 binding would disrupt enhancer–promoter interactions, as this might explain the strong effect of IBET treatment on transcription.

Strikingly, however, treatment with IBET had little or no effect on enhancer–promoter association. At the *MYC* enhancer, the major regions of contact with the promoter remained virtually unchanged by 90 min IBET treatment, with only small differences in interaction frequency (Fig. 3a, above). A similar result was

observed in the reciprocal Capture-C analysis from the enhancer, demonstrating only a subtle increase in interactions at the promoter (Fig. 3a, below). Even at a later 24 h timepoint, these enhancer–promoter interactions were mostly retained (Fig. 3a), suggesting that the looping structure is stable in the absence of high levels of BRD4. Strikingly, 24 h treatment with the BRD4 PROTAC molecule AT1 had a similarly minor effect on these enhancer–promoter interactions (Fig. 3b), arguing that maintenance of these contacts is unlikely to be mediated by the residual BRD4 remaining bound after IBET treatment.

While we do observe some rearrangement of the enhancer–promoter interactions after 24 h treatment, with an apparent shift from the distal to the more proximal region of the

**Fig. 2 IBET treatment results in large-scale transcriptional changes. a** Left: qRT-PCR analysis of gene expression following 1 µM IBET-151 treatment for the indicated times, using intronic PCR primers. Right: qRT-PCR analysis of gene expression using mature mRNA PCR primers. Values are normalized to *YWHAZ* mature mRNA levels, relative to DMSO treatment. Mean of three biological replicates; error bars show SEM. Source data are provided as a Source Data file. **b** MA plots for changes in nascent RNA levels following 90 min (left) or 24 h (middle) treatment with IBET. Right: correlation of $\log_2$ fold-change (logFC) of gene expression following IBET treatment for 90 min or 24 h. Statistically significant differences (red: increased; orange: decreased; gray: unchanged) from three biological replicates, FDR < 0.05. **c** Reference-normalized BRD4 ChIP-seq reads at BRD4 peaks from SEM cells treated with DMSO (*x*-axis) or IBET (*y*-axis) for 90 min. Solid line shows data trend (generalized additive model). **d** Metaplot of reference-normalized mean BRD4 levels at BRD4 peaks in SEM cells treated with DMSO (solid orange) or IBET (brown) for 90 min, or DMSO (dashed orange) or AT1 (black) for 24 h. Data are normalized to the peak DMSO read count for each treatment time. **e** qRT-PCR analysis of gene expression following IBET or AT1 treatment using mature mRNA PCR primers. Values are normalized to *YWHAZ* mature mRNA levels, relative to DMSO treatment. Mean of three (IBET treatments) or four (AT1 treatment) biological replicates; error bars show SEM. Source data are provided as a Source Data file. **f** Reference-normalized BRD4 and MED1 ChIP-seq at the *MYC* gene and enhancer and at *BCL2*. SEM cells were treated with DMSO or 1 µM IBET for 90 min, followed by BRD4 ChIP-seq, or with DMSO or 1 µM AT1 for 24 h, followed by BRD4 and MED1 ChIP-seq. **g** Reference-normalized BRD4 (orange) and MED1 (green) ChIP-seq reads at BRD4 peaks from SEM cells treated with DMSO (*x*-axis) or 1 µM AT1 (*y*-axis) for 24 h. Solid line shows data trend (generalized additive model).

enhancer with IBET (Fig. 3a), the broad interaction profile is maintained. It is worth noting that these changes are delayed relative to the early disruption of gene expression, as transcription of *MYC* was decreased after only 15 min IBET treatment and remained inhibited at 24 h (Fig. 2a, e).

We observed a similar maintenance of enhancer–promoter interactions at *BCL2*, where contact between the promoter and enhancer was preserved after even 24 h IBET or AT1 treatment (Fig. 3c, d) despite the decrease in transcription (Fig. 2a, e), arguing that although reduced BRD4 and Mediator binding can impact gene expression, these factors are not required for enhancer–promoter interactions at *MYC* or *BCL2*.

A more widespread analysis of Capture-C at a further 60 genes (Supplementary Data 1) demonstrated a similar response to IBET treatment, with minimal changes in promoter contacts (Fig. 3e, Supplementary Fig. 3a), whether they were up- or downregulated or transcriptionally unaffected by BET inhibition (Supplementary Fig. 3b). We also analyzed 30 of these genes (Supplementary Data 1) following treatment with AT1, finding that enhancer–promoter contacts were similarly unresponsive to BRD4 degradation (Fig. 3e, Supplementary Fig. 3c). This is in striking contrast to previous work from our lab showing a strong correlation between loss of enhancer–promoter interactions and reduction of transcription following DOT1L inhibition (DOT1Li), which results in decreased activity at H3K79me2/3-marked enhancers[30] (see Fig. 3e for comparison). Surprisingly, IBET and AT1 treatment led to small increases in interaction frequency at a number of genes (Fig. 3e, Supplementary Fig. 3a, c), despite the loss of BRD4. However, these changes were not always consistent between the two treatments (e.g., *CDK6*, Fig. 3e). In some cases, the increases correlated with a slight IBET-induced upregulation of transcription (e.g., *CDK6*, Fig. 3e), but in other cases it correlated with downregulation (e.g., *MBNL1*, Fig. 3e).

In order to quantify these differences, we measured the changes in interaction frequency at BRD4 peaks, reasoning that these sites were the most likely to be affected by loss of BRD4 binding. Because of the broad nature of the interaction profile, we used a 10 kb window centered on each peak (highlighted regions in Fig. 3e, Supplementary Fig. 3a, c). The majority of loci that showed statistical changes revealed a slight increase in interaction frequency following IBET or AT1 treatment (Fig. 3f; mean logFC = 0.11 and 0.22, respectively; Supplementary Data 3). This lack of a strong effect is not a limitation of the Capture-C technique, as DOT1Li-treated cells demonstrated a clear reduction in interaction frequency (Fig. 3e, f, mean logFC = −0.42, Supplementary Fig. 3d; Supplementary Data 3).

Longer treatment with IBET did not result in the delayed disruption of enhancer–promoter interactions (Supplementary Fig. 3e, Supplementary Data 3), as at *MYC* and *BCL2* (Fig. 3a, c).

Indeed, there was a clear correlation between the Capture-C changes observed at BRD4 peaks following 90 min or 24 h treatment with IBET (Supplementary Fig. 3f, R = 0.62), demonstrating that these chromatin structures are stable under conditions of reduced BRD4 and Mediator binding. The changes in interaction frequency also correlated, albeit more weakly, following 24 h IBET and AT1 treatment (Supplementary Fig. 3g, R = 0.42). Treatment with JQ1, which also disrupts the chromatin association of BRD4 (Supplementary Fig. 4a), resulted in similarly subtle effects on promoter interaction profiles (Supplementary Fig. 4b, Supplementary Data 3). Taken together, these results argue that while BRD4/Mediator binding may be required for enhancer function and gene transcription, they do not act primarily by stabilizing physical contact with the gene promoter.

While IBET and AT1 treatment result in the dissociation of BRD4 from chromatin, it is possible that other factors remain at enhancers that could facilitate promoter contact via low-affinity clustering interactions. It is also possible that the residual MED1 levels that remain at enhancers after IBET and AT1 treatment are sufficient to maintain enhancer–promoter interactions, despite the disruption of transcription. To induce a more generalized effect at these loci, we used 1,6-hexanediol, which is commonly employed to dissolve phase condensates[31,57,58]. Phase separation has recently been reported to be important for super-enhancer function[31,35–39]. Hexanediol treatment had a striking effect on global BRD4 and MED1 binding (Fig. 4a, Supplementary Fig. 4c), including at the *MYC* enhancers (Supplementary Fig. 4d). As has previously been observed[31], we found a strong reduction in BRD4 and MED1 association with super-enhancers compared to other enhancers (Fig. 4b). Hexanediol treatment also perturbed enhancer RNA (eRNA) transcription at super-enhancers (Fig. 4b), indicating a loss of super-enhancer function. Consistent with these results, we observed significant downregulation of genes associated with super-enhancers compared to other genes (Supplementary Fig. 4e). Hexanediol resulted in rapid changes in gene expression by nascent RNA-seq, with differential expression of more than 8000 genes after only 30 min (Fig. 4c, Supplementary Data 2).

Despite the strong downregulation of *MYC* and *BCL2* expression by hexanediol (Fig. 4d) and reduction in BRD4/MED1 enhancer binding (Fig. 4e, f), enhancer–promoter interactions at both genes were clearly retained (Fig. 4e, f). As with IBET and AT1 treatment, there were subtle rearrangements. Unlike with IBET treatment, there was a slight reduction in enhancer–promoter interactions at *BCL2* (Fig. 4f), but this is a minimal effect compared to enhancer–promoter reductions we have detected at other loci following DOT1Li[30] (Fig. 3e). Thus, using four different drug treatments (IBET, AT1, JQ1, hexanediol) to reduce BRD4 and Mediator binding at the *MYC* and

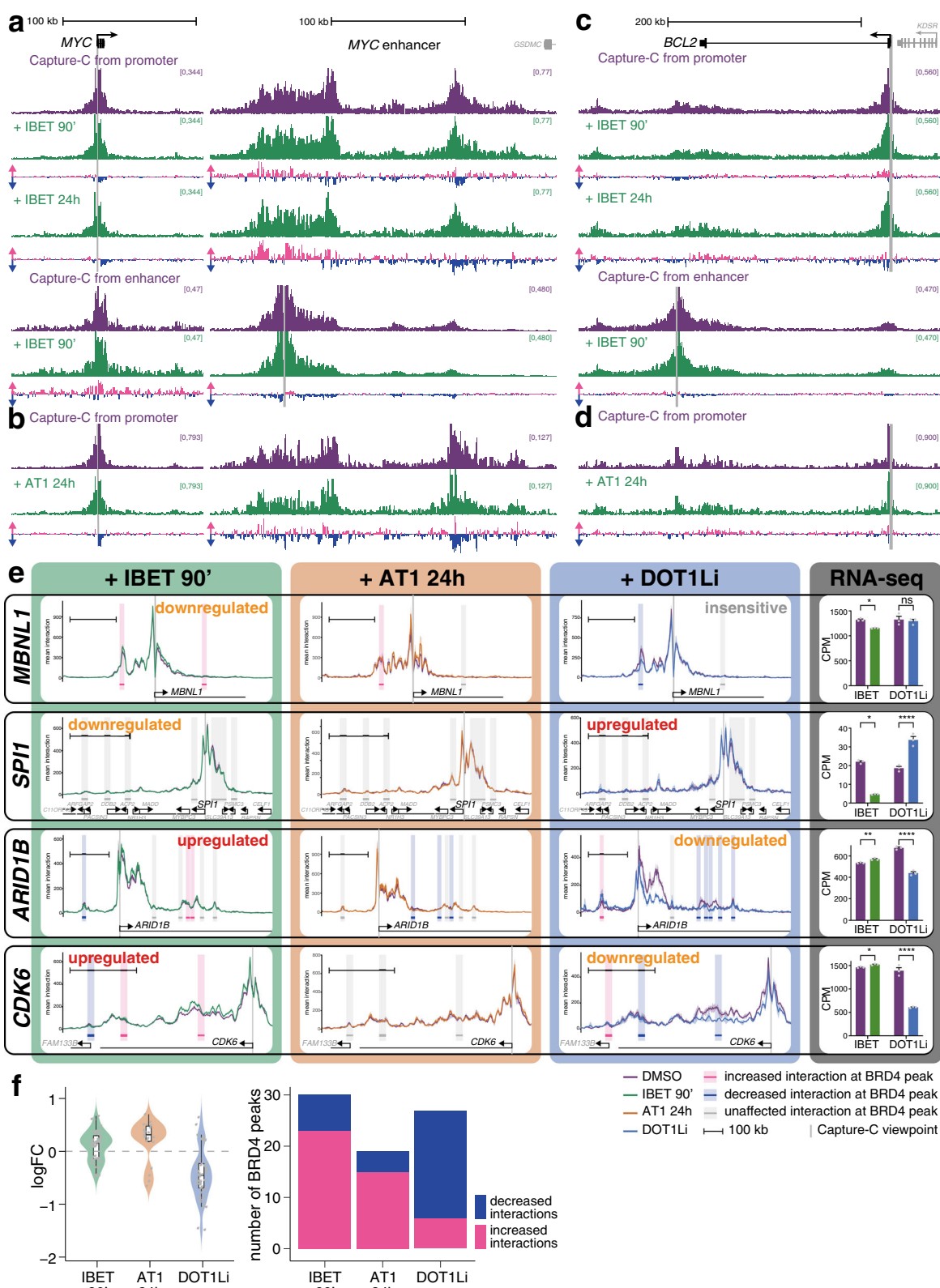

*BCL2* enhancers, we found very little evidence for a loss of interaction with their cognate promoters despite the reduction in transcription.

Analysis of other gene promoters by Capture-C revealed a similar lack of changes in interaction frequency following hexanediol treatment (Fig. 4g, Supplementary Data 3), regardless of the transcriptional change at the gene (Supplementary Fig. 4f).

Surprisingly, as with IBET treatment, the most common difference appeared to be a slight increase in contact frequency (Fig. 4g; mean logFC = 0.18). Notably, statistical analysis identified a clear correlation in the effects observed at BRD4 peaks for the five treatments used (IBET for 90 min or 24 h, AT1 for 24 h, JQ1 for 90 min and hexanediol for 30 min), and anticorrelation with DOT1L inhibition (Supplementary Fig. 4g).

**Fig. 3 BET inhibition has minimal effects on enhancer–promoter interactions. a** Capture-C from the *MYC* promoter (above) or enhancer (below) following 90 min DMSO treatment (purple) or 90 min or 24 h 1 µM IBET treatment (green). Only the promoter and enhancer regions are shown. Differential tracks show the change in profile in IBET-treated samples compared to DMSO treatment for the same time period: pink bars show increases; blue bars show decreases. Mean of three biological replicates. **b** Capture-C from the *MYC* promoter following 24 h DMSO (purple) or 1 µM AT1 treatment (green), as in (a). **c**, **d** Capture-C from the *BCL2* promoter or enhancer, as in **a**, **b**. **e** Capture-C traces from gene promoters following treatment with DMSO (purple line), IBET for 90 min (left, green line), AT1 for 24 h (middle left, orange line) or EPZ-5676 for 7d (DOT1Li, middle right, blue line). Ribbon shows ±1 SD for three replicates. Vertical gray bar indicates the capture point for each gene. Horizontal bars show 10 kb region around BRD4 ChIP-seq peaks. Shading highlights effect of IBET treatment on promoter interaction frequency within that window: pink bars indicate statistically significant increases; blue bars indicate decreases; gray bars indicate no significant difference (Holm–Bonferroni adjusted *p*-value < 0.05, paired Mann–Whitney test; adjusted *P*-values are given in Supplementary Data 3). Scale bars show 100 kb. Transcriptional effect of the drug treatment on the gene is indicated. Right: Nascent RNA-seq levels for each gene under control or indicated treatment conditions. ****FDR < 0.0001, **FDR < 0.01, *FDR < 0.05, ns no significant change; FDR values are given in Supplementary Data 2. Mean of three biological replicates; error bars show SEM. DOT1Li Capture-C and RNA-seq data are taken from[30]. **f** Left: change in interaction frequency (mean logFC of three replicates) between promoters and BRD4 peaks (10 kb windows) of significantly affected (Holm–Bonferroni adjusted *p*-value < 0.05, paired Mann–Whitney test; adjusted *P*-values are given in Supplementary Data 3) interactions following 90 min IBET, 24 h AT1, or 7d DOT1Li treatment. Nonsignificantly affected interactions are not plotted. Violin plot shows frequency distribution. Boxplot midline shows median, with upper and lower hinges showing 25th and 75th percentile, respectively. Upper and lower hinges extend to the largest and smallest datapoints within 1.5 times the interquartile range of either hinge. Dots represent individual BRD4 peaks. Source data are provided as a Source Data file. Right: number of BRD4 peaks (10 kb windows) that show statistically significant increased (pink) or decreased (blue) interactions following 90 min IBET, 24 h AT1, or 7d DOT1Li treatment.

Thus, four distinct drug treatments disrupting BRD4 localization produced a similarly weak effect on enhancer–promoter interactions at the genes studied. From this we conclude that depletion of BRD4 and MED1 binding at enhancers can have a strong impact on transcription, but this is not sufficient to disrupt enhancer–promoter interactions. This contrasts strongly with our past work where loss of H3K79me2/3 at KEEs causes both decreased transcription and reduced enhancer–promoter interactions[30] (Fig. 3e, f).

**Cohesin/CTCF binding patterns support a role in mediating a subset of enhancer–promoter interactions.** Another mechanism that has been proposed to mediate interaction between promoters and enhancers is the loop extrusion model, which is also used to explain the generation of higher-order chromatin structures. In this model, a loop of chromatin is fed through cohesin, until it is paused by two CTCF molecules bound in a convergent orientation[9,10]. In SEM cells, CTCF and RAD21 show a strong positive correlation at ATAC peaks (Fig. 1b), suggesting that all or most of these CTCF binding sites are competent to enrich or stabilize RAD21 association with chromatin. It is unclear whether loop extrusion may contribute to the increased local interactions between promoters and enhancers, although recent work has suggested this possibility[21,22,59]. In support of this idea, CTCF and RAD21 can be observed at many enhancers and promoters, as well as non-enhancer/promoter ATAC peaks in SEM cells (Fig. 1a). Further, the binding of CTCF or RAD21 at the *MYC* or *BCL2* promoter or enhancer is mostly unperturbed by IBET or hexanediol treatment (Supplementary Fig. 5a, b), correlating with the maintenance of enhancer–promoter interactions.

To investigate whether cohesin/CTCF binding is a plausible mechanism to mediate enhancer–promoter contact, we compared the ChIP-seq profiles of these proteins to our Capture-C promoter interaction profiles. At *MYC* both the promoter and enhancer regions are associated with several closely spaced CTCF/RAD21 peaks (Fig. 5a). Strikingly, the promoter CTCF-bound motifs are oriented towards the enhancers, and the enhancer binding sites are oriented towards the promoter (Fig. 5a, blue triangles), suggesting that any pairing of these would produce a convergent CTCF dimer, consistent with cohesin-mediated DNA looping[10,60,61]. Indeed, the promoter CTCF sites have previously been shown to play a role in interaction with

distinct enhancer regions in different cancer cell lines[23]. At the *MYC* enhancer, the proximal domain is bounded by a pair of CTCF/RAD21 binding sites, and the major peak of the distal region is marked by two binding sites (Fig. 5a). Given that these CTCF/RAD21 sites all overlap with key points in the promoter interaction profile, this indicates that there may be multiple opportunities to stabilize contacts with the promoter via this mechanism.

As at *MYC*, there are multiple CTCF/RAD21 peaks at the promoter of *BCL2*, and a clear convergent peak at the distal interaction region (Fig. 5b; overlapping with the *BCL2* enh 2 primer pair), which is not marked with BET proteins or other enhancer features (Fig. 1d). There are also two CTCF sites (convergent with the promoter), which overlap with the broad interaction domain at the enhancer, although notably these CTCF sites occupy a distinct region to the peak of BRD4 and do not fully correlate with the interacting region. This suggests that additional or alternative factors to CTCF may facilitate contact between the *BCL2* enhancer and promoter.

We expanded this analysis to include other enhancer-associated genes for which we had promoter interaction data. Many RAD21/CTCF peaks within the analyzed domains were not associated with interactions with the target promoter (e.g., in the region between the *MYC* promoter and enhancer, Fig. 5a, upper), but may be involved in mediating other DNA–DNA contacts. However, we identified numerous instances of promoter-oriented RAD21/CTCF peaks overlapping with enhancer–promoter interactions (Fig. 5c, pink highlights).

Our data suggest that at a subset of enhancers, CTCF and cohesin may be partly responsible for facilitating enhancer–promoter interactions. Recent work in SEM cells, the cell line studied here, used Capture-C following CTCF degradation to test the effect of CTCF loss on the interaction between the *MYC* promoter and enhancer[22]. Consistent with the model that CTCF binding stabilizes enhancer–promoter interactions via cohesin-mediated loop extrusion, loss of CTCF was found to reduce both *MYC* expression and interactions between the *MYC* enhancer and promoter[22]. However, reanalysis of these data using the same approach as for our Capture-C contrasts the dramatic decreases in interaction observed with CTCF degradation with the minor changes observed following BRD4 degradation with AT1 (Fig. 5d). These results argue that, at least at *MYC*, the loop extrusion model can explain enhancer–promoter contact and may be important for gene expression.

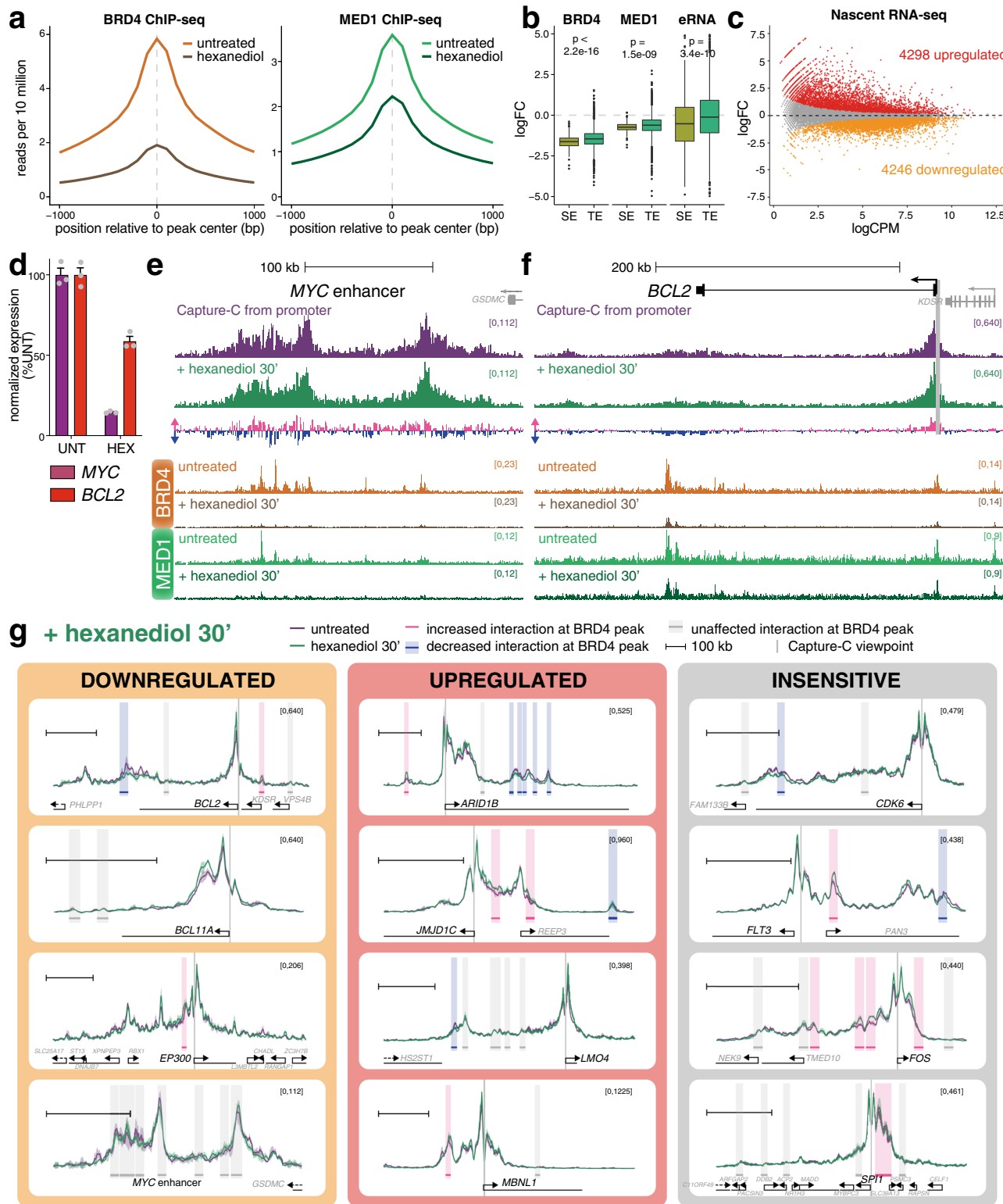

## Discussion

Maintenance of enhancer–promoter looping is thought to be crucial for gene activation, but emerging evidence and the data presented here suggest that enhancer–promoter contact and gene activation may be partially separable events. Importantly, physical interaction with the promoter may not be necessary for all enhancers[5,6], although it appears to be a requisite for most. At the same time, enhancer–promoter looping alone is not sufficient for activation, as enhancer–promoter contacts have been observed in the absence of

transcription[62–64]. We show here that transcription can be disrupted with minimal changes in enhancer–promoter interaction frequency, as has been observed at the β-globin locus[65]. In contrast, artificially stabilizing enhancer–promoter loops can activate transcription[24,66–70], indicating that the conversion of unproductive enhancer–promoter contacts to a functional complex may be dependent on the presence of additional factors.

These data suggest a model whereby stabilization of enhancer–promoter loops is a necessary but not sufficient

**Fig. 4 Dissolution of phase condensate structures with 1,6-hexanediol does not perturb enhancer–promoter interactions. a** Metaplot of reference-normalized mean BRD4 and MED1 levels at BRD4 peaks in untreated SEM cells (light color) or cells treated with 1.5% 1,6-hexanediol for 30 min (dark color). **b** Boxplot showing the $log_2$ fold-change (logFC) in reference-normalized levels of BRD4 and MED1 and nascent RNA at super-enhancers (SE; olive), or typical enhancers (TE; green) following treatment with 1.5% 1,6-hexanediol for 30 min. Nascent RNA (eRNA) was measured over 1 kb windows centered on intergenic ATAC-seq peaks overlapping with SEs and TEs. $p$ values indicate the statistical significance of the difference in logFC between SEs and TEs (Wilcoxon rank sum test; for BRD4, $p < 2.2 \times 10^{-16}$; MED1, $p = 1.5 \times 10^{-9}$; eRNA, $p = 3.4 \times 10^{-10}$). Midline shows median, with upper and lower hinges showing 25th and 75th percentile, respectively. Upper and lower hinges extend to the largest and smallest datapoints within 1.5 times the interquartile range of either hinge; outliers are plotted as dots. Analysis of one experiment (BRD4 and MED1 ChIP-seq) or three independent experiments (eRNA). **c** MA plot of changes in nascent RNA levels following 30 min treatment with 1.5% 1,6-hexanediol. Mean of three biological replicates. Statistically significant differences (red: increased; orange: decreased; gray: unchanged) from three biological replicates, FDR < 0.05. **d** Quantification of *MYC* and *BCL2* nascent RNA-seq levels in untreated SEM cells or cells treated with 1.5% 1,6-hexanediol for 30 min. Mean of three biological replicates, normalized to expression in untreated cells; error bars show SEM. Source data are provided as a Source Data file. **e** Capture-C from the *MYC* promoter from untreated SEM cells (purple) or following 30 min treatment with 1.5% 1,6-hexanediol (green), mean of three biological replicates. Differential tracks show the change in profile in hexanediol-treated samples: pink bars show increases; blue bars show decreases. Reference-normalized BRD4 and MED1 ChIP-seq from untreated SEM cells and cells treated with 1,6-hexanediol for 30 min. Only the enhancer region is shown. **f** Capture-C from the *BCL2* promoter and reference-normalized ChIP-seq, as in **e**. **g** Capture-C traces at genes that are transcriptionally downregulated (orange), upregulated (red) or unaffected (gray) by 30 min 1,6-hexanediol treatment. Purple line shows the profile in untreated cells; green line is from hexanediol-treated cells; ribbon shows ±1 SD for three replicates. Vertical gray bar indicates the capture point for each gene. Horizontal bars show 10 kb region around BRD4 ChIP-seq peaks. Shading highlights effect of hexanediol treatment on promoter interaction frequency within that window: pink bars indicate statistically significant increases; blue bars indicate decreases; gray bars indicate no significant difference (Holm–Bonferroni adjusted $p$-value < 0.05, paired Mann–Whitney test; adjusted $P$-values are given in Supplementary Data 3). Scale bar shows 100 kb.

precondition for gene activation, and the protein complexes that facilitate looping are not sufficient to directly promote gene expression. A distinct, functionally separable, stage of gene activation follows, where enhancer-associated factors interact with the promoter, producing transcriptional upregulation. Indeed, enhancer–promoter loop structures likely create an opportunity for contact between factors at these two loci. A number of enhancer-associated factors, including BRD4 and Mediator, have been observed to form phase-separated condensates in vivo[31,35], and we found that disruption of BRD4 and Mediator binding had appreciable effects on transcription. This suggests that there may be a role for the low-affinity interactions that can drive phase condensation in facilitating functional interactions between enhancers and promoters, particularly at regions of high activator density, such as super-enhancers, for example by stabilizing the binding of RNA polymerase at the promoter[35,39,71,72].

Recent models have proposed that the low-affinity interactions that drive phase condensation may be sufficient for both enhancer–promoter colocalization as well as promoting transcriptional activation[37,41,73,74]. Computational simulation has suggested that formation of these condensates may promote long-range chromatin interactions[37,41], and BRD4 is capable of driving clustering of acetylated chromatin in vitro[75]. In support of this, BRD4 intrinsically disordered regions (IDRs) targeted to telomeric sequences appear to bring loci together[42], and dissolution of phase condensates prevents the estrogen-induced colocalization of enhancers[76]. However, it is unclear how closely these observations represent physiological enhancer–promoter interactions, or whether these results are representative of mechanisms functioning generally at most enhancers. Our data demonstrate that these low-affinity interactions are not necessary for the maintenance of enhancer–promoter contacts, as loss of BRD4 chromatin binding had no effect on promoter interaction profiles. A similar lack of effects was recently observed, albeit at lower resolution by Hi-C, in Mediator mutant mouse embryonic stem cells (mESCs)[21]. Importantly, the high resolution and sensitivity of Capture-C confirms the lack of even subtle changes in enhancer–promoter contacts in our experiments, for example localized to BRD4 binding sites.

We note that, although our drug treatments reduced BRD4 and Mediator binding, some signal was still detectable by ChIP. This raises the possibility that high levels of BET/Mediator are needed for transcription, but low levels of BET/Mediator binding may be sufficient to maintain enhancer–promoter interactions (Fig. 6). Arguing against this model, BRD4 degradation, which produced a much stronger reduction in BRD4 and MED1 levels, was no more effective at disrupting enhancer–promoter interactions than BET inhibition. The most common change we observed when BET/Mediator binding was reduced was actually a slight increase in enhancer–promoter interaction frequencies. This behavior is similar to that observed in a Mediator mutant cell line[21], suggesting that it may be a genuine consequence of Mediator loss. It is possible that these structural changes are an indirect effect of the transcriptional disruption following BRD4/Mediator loss, as has been observed before[77,78], although there was no correlation with transcriptional response.

What, then, is responsible for establishing and maintaining enhancer–promoter interactions? We favor a role for cohesin and CTCF at the subset of genes where they are bound, potentially in combination with low levels of BRD4 and Mediator (Fig. 6) and other activators such as transcription factors. Enhancer–promoter proximity could therefore be the result of an aggregate of interactions, partially stabilized by cohesin. While the loop extrusion model is widely accepted in the maintenance of higher-order domain structure, a function at enhancers is less clear[73]. Depletion of cohesin or its loader NIPBL disrupts interactions between some promoters and distal enhancers[21,59,79,80], although in some cases this may be an indirect effect through loss of TAD boundaries rather than physical contact with enhancers. As has been reported[23,59], in our analysis we observed an enrichment for CTCF/RAD21 binding at promoter-interacting loci, consistent with a direct role for loop extrusion in mediating enhancer–promoter interactions. Indeed, loss of Rad21 results in decreased enhancer–promoter contacts and transcriptional downregulation at *Sik1* and *Elf3* in mESCs[79]. Similarly, enhancer–promoter interactions at *MYC* are dependent on CTCF[22,23]. However, it is possible that this gene is unusual, as long-distance enhancer–promoter interactions over 1 Mb are not common. Strikingly, the majority of genes rapidly downregulated following CTCF degradation in mESCs show CTCF binding at the promoter, although this effect was proposed to be a result of a looping-independent function of CTCF in transcription[81].

One complicating aspect of the role for cohesin in enhancer–promoter contacts is the fact that disruption of loop

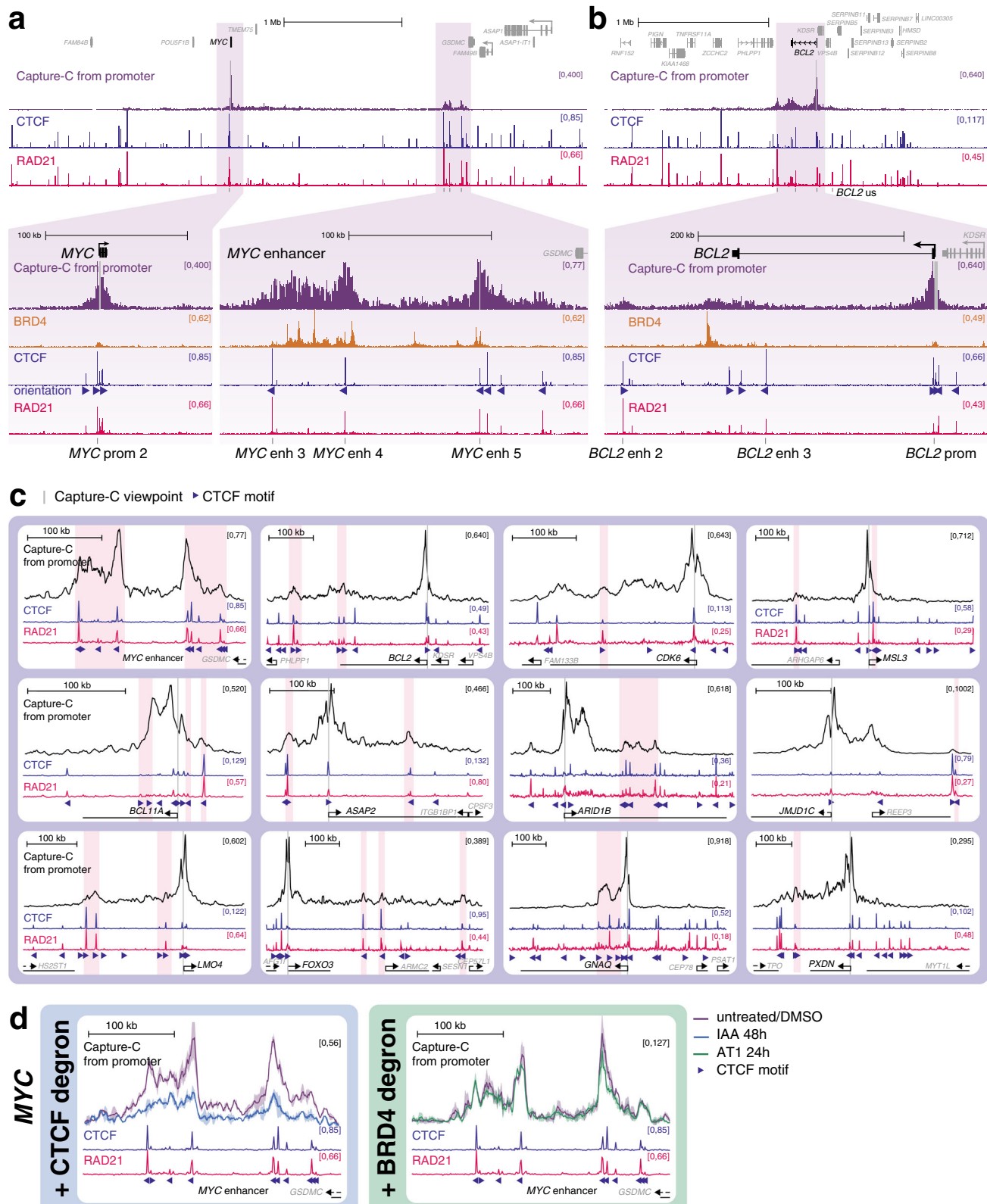

extrusion, either by loss of cohesin itself, its loader NIPBL or CTCF, does not have widespread effects on gene expression[22,59,80–83], although cohesin may be important for inducible gene expression[84]. This suggests, assuming that the majority of enhancer–promoter interactions are productive, that cohesin is not essential for most enhancer function. However, given that our current understanding of enhancer function remains incomplete, this point alone is insufficient to rule out a role for loop extrusion in linking at least a subset of enhancers to promoters.

It is likely that multiple mechanisms exist to facilitate enhancer–promoter interactions at different genes, and may function at least partly redundantly. Recent work using Promoter Capture Hi-C has shown that many, but not all, promoter interactions are unaffected by cohesin or CTCF depletion,

**Fig. 5 CTCF and RAD21 may be responsible for mediating enhancer–promoter interactions at *MYC* and *BCL2*. a** Capture-C and ChIP-seq for BRD4, CTCF and RAD21 at the *MYC* gene and enhancer region. Capture-C was conducted using the *MYC* promoter as the viewpoint, indicated by a vertical gray bar, mean of three biological replicates. Orientation of CTCF motifs at peaks is indicated by triangles. Locations of primers used for CTCF/RAD21 ChIP-qPCR (see Supplementary Fig. 5) are shown at the bottom of the figure. **b** Capture-C and ChIP-seq data at *BCL2*, as in **a**. **c** Capture-C traces from untreated cells (black line; mean of three replicates) and ChIP-seq for CTCF (blue) and RAD21 (pink). Vertical gray bar indicates the capture point for each gene. Orientation of CTCF motifs at peaks is indicated by triangles. Scale bars show 100 kb. Pink shading highlights promoter-interacting regions that overlap with CTCF/RAD21 peaks (visually determined). **d** Left: Capture-C profile from the *MYC* promoter showing the *MYC* enhancer in SEM cells with AID-tagged CTCF, either untreated (purple line) or treated with indole-3-acetic acid (IAA) for 48 h (blue line), which targets CTCF for degradation. Data are replotted from[22], mean of two independent clones. Right: Capture-C profile from the *MYC* promoter showing the *MYC* enhancer in SEM cells treated with DMSO (purple line) or AT1 (green line) for 24 h, mean of three biological replicates. CTCF (blue) and RAD21 (pink) ChIP-seq tracks and CTCF motif orientations (triangles) are shown.

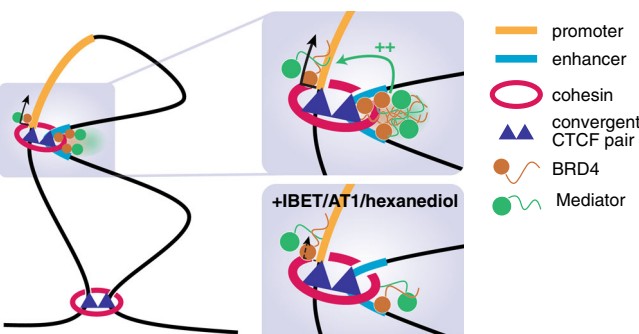

**Fig. 6 Model for enhancer–promoter interaction.** Higher-order chromatin boundaries are maintained by cohesin loops associated with convergent CTCF dimers. Within a domain, many enhancer–promoter contacts are associated with RAD21/CTCF peaks, and we suggest that similar cohesin loops are required for a subset of these interactions. At some enhancers (for example super-enhancers) a high concentration of factors such as BRD4 and Mediator drive the formation of phase condensates, and these may increase interactions with factors at the promoter, held nearby by cohesin loops. These interactions may be required to activate or increase transcription from the promoter. Upon addition of IBET, AT1, or 1,6-hexanediol, BRD4 and Mediator binding is reduced at the enhancer and phase condensates are dissolved, disrupting interaction with factors at the promoter. This disrupts gene expression, but does not affect enhancer–promoter proximity as the two loci remain held together by other factors, for example cohesin.

indicating that loop extrusion likely has a role in only a subset of these contacts[59]. For example, deletion of the sole CTCF site in the *Sox2* super-enhancer in mESCs reduces, but does not abolish, contact with the promoter[60]. Indeed, in our analysis, while many of the promoter interaction sites overlap with correctly oriented CTCF sites, there are also many sites of interaction that lack an obvious peak of CTCF binding (e.g., at *ARID1B*). We also observe broad regions of interaction that are bookended by peaks of CTCF/RAD21 (e.g., at the *MYC* enhancer), which suggests that, while CTCF and cohesin may define the borders of interaction, additional factors may play a role in more local contacts with the promoter. H3K4me1, a mark of enhancers, has itself been found to interact with cohesin[29]. Transcription factors are another plausible anchor, with a number of mechanisms proposed[85]. For example, degradation of Oct4 (Pou5f1), but not Nanog, in mESCs results in a loss of Rad21 association with TF binding sites[79], arguing that specific TFs may be able to recruit or stabilize cohesin, potentially directing enhancer–promoter interactions. Mediator itself has been suggested to physically interact with cohesin[86]. Additional structural proteins may also be important, for example YY1[24] and WIZ[87].

There may also be a role for noncoding RNA in enhancer–promoter interactions[25,26,88–92]. Notably, while mRNA has been shown to direct the formation of phase condensate compartments in the cytoplasm[93] and eRNAs have been proposed to play a similar role in the formation of enhancer–promoter complexes[25,26] our results following dissolution of phase condensates with hexanediol treatment argue that these interactions are not sufficient for maintaining contact at many genes. However, RNA may play other roles in directing enhancer–promoter interactions, for example recruitment of cohesin[89,90], or the process of enhancer transcription itself may be important[91].

Our results show that BRD4 and Mediator play a key role in the transcription of many genes, but they achieve this mainly via a functional rather than structural role. The high levels of these proteins at enhancers relative to promoters argues that contact between these loci is likely important for expression of many genes, but that this interaction functions primarily to enrich the local concentration of enhancer-bound factors at the promoter. Similarly, while the presence of phase condensates appears to be important for the transcription of super-enhancer-associated genes, this is likely a mechanism to concentrate key transcription-related proteins at the enhancer–promoter complex, and inhibition of this clustering has little or no effect on enhancer–promoter looping. Physical contact between promoter and enhancer is not, per se, sufficient for transcription, and is not dependent on high levels of BRD4 or Mediator.

## Methods

**Cell culture and cell lines**. SEM (an MLL-AF4 B-ALL cell line)[94] cells were purchased from DSMZ (www.cell-lines.de) and cultured in IMDM with 10% FBS and Glutamax. For drug treatments cells were diluted to $5 \times 10^5$ cells/ml. IBET-151 (Tocris) was used at a final concentration of 1 μM, AT1 (Tocris) at 1 μM, (+)-JQ1 (Tocris) at 1 μM and 1,6-hexanediol (Merck) at 1.5% (w/v).

**Chromatin immunoprecipitation**. Briefly, double-fixed samples (2 mM disuccinimidyl glutarate (Sigma) for 30 min followed by 1% formaldehyde (Sigma) for 30 min) were sonicated in batches of $10^7$ cells using a Covaris (Woburn, MA) following the manufacturer's recommendations. Antibodies used for ChIP are detailed in Supplementary Table 1. Antibody-chromatin complexes were isolated using a 1:1 mixture of magnetic Protein A- and Protein G-dynabeads (Thermo-Fisher Scientific) and washed three times with a solution of 50 mM HEPES-KOH, pH 7.6, 500 mM LiCl, 1 mM EDTA, 1% NP-40, and 0.7% Na deoxycholate. Following a Tris-EDTA wash, samples were eluted with 50 mM Tris-HCl, pH 8.0, 10 mM EDTA and 1% SDS, then treated with RNase A and proteinase K. DNA was purified using a PCR purification kit (Qiagen). For ChIP-qPCR, DNA was quantified relative to input chromatin, using primers listed in Supplementary Table 2. For ChIP-seq, DNA libraries were generated using the NEBNext Ultra II DNA Library Preparation kit (NEB). Samples were sequenced by 40 bp paired-end sequencing using a NextSeq 500 (Illumina).

**ChIP-seq bioinformatic analysis**. Quality control of FASTQ reads, alignments, PCR duplicate filtering, blacklisted region filtering and UCSC data hub generation were performed using the NGseqBasic pipeline[95]. Briefly, QC was checked with fastQC (http://www.bioinformatics.babraham.ac.uk/projects/fastqc/), then reads were mapped against the human genome assembly (hg19) using Bowtie[96]. Unmapped reads were trimmed with trim_galore (https://www.bioinformatics.babraham.ac.uk/projects/trim_galore/) and remapped. Short unmapped reads from this step were combined using Flash and mapped again. PCR duplicates were removed with samtools rmdup[97], and any reads mapping to Duke blacklisted

regions (UCSC) were removed with bedtools. Sequence tag (read) directories were generated from the sam files with the Homer tool makeTagDirectory[98]. The command makeBigWig.pl was used to generate bigwig files for visualization in UCSC, normalizing tag counts to tags per $10^7$. Peaks were called using the Homer tool findPeaks, with the input track provided for background correction, using -style histone or -style factor options to call peaks in histone modification or transcription factor datasets, respectively. Super-enhancers were identified using findPeaks with the options -style super -minDist 12500 -L 1, providing tag directories for H3K27ac, H3K4me1, BRD4, and MED1. Typical enhancers were defined as H3K27ac peaks that overlap with H3K4me1 peaks, are >1 kb from an annotated TSS and do not overlap with a super-enhancer. Genes were annotated by assigning enhancers to the nearest TSS. Metagene profiles were generated using the Homer tool annotatePeaks.pl. Heatmaps were drawn using the R package heatmap3. CTCF motif orientations were assigned using the FIMO function of the MEME Suite[99].

**Reference-normalized ChIP-seq**. Reference normalization[100] was achieved by adding fixed *Drosophila melanogaster* S2 cells to fixed SEM cells at the ChIP lysis step prior to sonication, in a 1:4 ratio, and the ChIP protocol was followed as normal. After sequencing, input and IP reads were mapped to both hg19 and dm3 genome builds, and hg19 read counts were adjusted based on the ratio of dm3:hg19 reads in input and IP control and treatment samples.

**qRT-PCR**. Total RNA was extracted and DNase I-treated from $10^6$ cell pellets using the RNeasy Mini kit (Qiagen). RNA was reverse-transcribed using SuperScript III (ThermoFisher Scientific) with random hexamer primers, then quantified using SyBr Green or TaqMan qPCR (see Supplementary Table 2 for primers). Gene expression was normalized to mature mRNA levels of the housekeeping gene *YWHAZ*.

**Nascent RNA-seq**. $10^8$ SEM cells at $5 \times 10^5$ cells/ml were treated with 500 μM 4-thiouridine (4-SU) for the final 1 h (IBET treatments) or 30 min (hexanediol treatment) of the drug treatment time (e.g., 30 min IBET treatment before 4-SU addition for 1 h, giving 90 min total IBET treatment time). Pelleted cells were lysed with Trizol (ThermoFisher Scientific) and total RNA was precipitated and DNase I-treated. 4-SU-incorporated RNA was biotinylated with EZ-link Biotin-HPDP (ThermoFisher Scientific) and purified with Streptavidin bead pull-down (Miltenyi). DNA libraries were generated from RNA using the NEBNext Ultra Directional RNA Library Preparation kit (NEB). Samples were sequenced by 75 bp paired-end sequencing using a NextSeq 500 (Illumina).

**RNA-seq bioinformatic analysis**. Following QC analysis with fastQC (http://www.bioinformatics.babraham.ac.uk/projects/fastqc) reads were aligned against the human genome assembly (hg19) using STAR[101]. Duplicate reads were removed using the picard command MarkDuplicates.jar (http://broadinstitute.github.io/picard). Gene expression levels were quantified as read counts using the featureCounts function from the Subread package with default parameters[102]. The read counts were used to identify differential gene expression between conditions and generate RPKM values using the edgeR package[103]. Genes were considered differentially expressed if they had an adjusted *p*-value (FDR) of less than 0.05. Strand-specific RNA-seq was visualized on UCSC using the bam file as input for Homer[98] commands makeTagDirectory (with options -flip and -sspe) and makeMultiWigHub.pl (with option -strand separate).

**Capture-C**. Next-generation Capture-C was performed as described[18]. Briefly, $2 \times 10^7$ fixed SEM cell nuclei were digested with DpnII and used to generate a 3 C library. Libraries were sonicated to a fragment size of 200 bp and Illumina paired-end sequencing adaptors (NEB) were added, using Herculase II (Agilent) to amplify the DNA. Indexing was performed in duplicate to maintain library complexity, with libraries pooled after indexing. Previously-designed Capture-C probes[30] targeting promoters or enhancers (Supplementary Data 1) were used to enrich for target sequences with two successive rounds of hybridization, streptavidin bead pull-down (ThermoFisher Scientific), bead washes (Nimblegen SeqCap EZ) and PCR amplification (NimbleGen SeqCap EZ accessory kit v2). Captured DNA was sequenced by 150 bp paired-end sequencing using a NextSeq 500 (Illumina). Data analysis was performed using scripts available at https://github.com/Hughes-Genome-Group/CCseqBasicF/releases. Capture-C promoter interactions overlapping with indicated ChIP-seq/ATAC-seq peaks were quantified for statistical analysis. Peaks outside of the bounds of Capture-C interaction domains (visually determined using UCSC genome browser) and those on trans chromosomes were removed from the analysis. Peaks within 10 kb of the Capture-C probe hybridization site were also removed. Holm–Bonferroni adjusted *p*-values for each peak were calculated by comparing all of the normalized read counts for each DpnII fragment and all replicates within a peak using a paired Mann–Whitney test for the two treatment conditions.

**Western blotting**. Salt-soluble proteins were extracted from $1 \times 10^6$ SEM cells by incubating cells in a high-salt lysis buffer (20 mM Tris-HCl pH 8.0, 300 mM KCl,

5 mM EDTA, 20% glycerol, 0.5% IGEPAL CA-630, protease inhibitor cocktail), and protein levels were analyzed by western blotting[104]. Antibodies used are detailed in Supplementary Table 1.

**Statistical analysis**. Statistical analyses used and sample sizes are indicated in figure legends; n numbers refer to independent experiments. All tests were conducted two-tailed, all correlation analyses were conducted using the Pearson method.

**Reporting summary**. Further information on research design is available in the Nature Research Reporting Summary linked to this article.

## Data availability
All sequencing data that support the findings of this study have been deposited in the Gene Expression Omnibus (GEO) with the accession code GSE139437. GEO accession numbers for datasets used from previous publications can be found in Supplementary Table 3. All other relevant data supporting the key findings of this study are available within the article and its Supplementary Information files or from the corresponding author upon reasonable request. A reporting summary for this Article is available as a Supplementary Information file. Source data are provided with this paper.

## Code availability
ChIP-seq and ATAC-seq data were analyzed using the NGSeqBasic pipeline[95]. Capture-C data analysis was performed using scripts available at https://github.com/Hughes-Genome-Group/CCseqBasicF/releases.

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

## Acknowledgements

T.A.M., N.T.C., E.B., L.G., R.T., J.K., and M.T. were funded by Medical Research Council (MRC, UK) Molecular Haematology Unit grant MC_UU_12009/6 and MC_UU_00016/6. P.H. and J.O.J.D. are funded by an MRC Clinician Scientist Award to J.O.J.D. (MR/R008108). P.F. was supported by the Medical Research Council (MR/N010051/1). We would like to thank Jill Brown for her comments on the manuscript.

## Author contributions

N.T.C., E.B., and T.A.M. conceived the experimental design; N.T.C., E.B., L.G., T.A.M., J.K., and M.T. carried out experiments; P.F. provided reagents; N.T.C., R.T., and E.R. analyzed and curated the data; N.T.C., E.B., and T.A.M. interpreted the data; P.H., C.L., and J.O.J.D. provided expertise; N.T.C. and T.A.M. wrote the manuscript; all authors contributed to reviewing and editing the manuscript; T.A.M. provided supervision and funding.

## Competing interests

T.A.M. is a founding shareholder of OxStem Oncology (OSO), a subsidiary company of OxStem Ltd. J.O.J.D. is a co-founder of Nucleome Therapeutics Ltd. to which he provides consultancy. All other authors have no competing interests.
