## [Peer Review File · Nature Communications]

Reviewers' comments:

Reviewer #1 (Remarks to the Author):

The authors investigated the effect of the inhibition of BET proteins on transcription and enhancer-promoter interactions. They found that in leukemia cells, the ubiquitous co-factors BRD4 and Mediator are enriched at both enhancers and promoters. Treatment of leukemia cells with the BET inhibitor iBET-151 lead to reduced BRD4 binding, rapid decrease of nascent transcription, but a negligible effect on enhancer promoter interactions measured by Capture C. Furthermore, treatment of the cells with 1,6-hexanediol also had negligible effect on enhancer-promoter interactions in contrast to a Dot1l inhibitor.

This paper contributes to a growing body of work that ubiquitous transcriptional co-factors such as BRD4 and Mediator have key functions in controlling transcriptional activity, but have little contribution to interactions between promoters and enhancers [see e.g. (El Khattabi et al., 2019)].

The experiments appear thoroughly performed and analyzed. I have concerns regarding the effect of iBET-151 and the interpretation of the data, the conflict of some of the authors' terminology with the existing literature, and referencing of key statements in the manuscript.

Specific comments

1. The authors investigated the effect of the inhibition of BET proteins on transcription and enhancer promoter interactions. The key experiment was treating the leukemia cells with the BET inhibitor iBET-151, which lead to a negligible effect on enhancer promoter interactions as assayed by Capture C for about 60 loci. The key concern with this experiment is that the treatment regime of the authors appears to have only a moderate effect on BRD4 binding to the genome, and the majority of the BRD4 ChIP-Seq signal persists after the inhibitor treatment at key loci the authors analyzed (Figure 2f). The results simply do not support the authors' claim that "Loss of BET and Mediator binding is associated with large transcriptional changes at key oncogenic gene targets" (page 7, line 26), although the statement may be ultimately correct [predominantly based on the existing literature, e.g. (El Khattabi et al., 2019)]. The results thus need to be consistently described in the text along the lines of "BET inhibition is associated with large transcriptional changes at key oncogenic gene targets."

2. The authors try to instigate that the previously reported ability of Mediator and BRD4 to form phase-separated condensates is linked to the transcriptional effects they observe (e.g. page 9), but this link is not supported by data. For example, the authors write "Consistent with the loss of MED1 foci in cells following BET inhibition³⁴,..." (page 9, line 5), and cite the recent paper from the Cisse lab (Cho et al., 2018). First, the Cisse lab used different cells (murine embryonic stem cells), used a different inhibitor (JQ1), used a sophisticated super-resolution-based imaging method to detect condensates in live cells, and did not demonstrate displacement of BRD4 from the genome after their treatment regime. If the authors want to claim any link between condensate formation and their results on transcription, they need to at least perform BRD4 and Mediator immunofluorescence, and show that in the leukemia cells the iBET-151 treatment dissolves BRD4/Mediator foci. Further supporting evidence the authors can quickly collect here is sequencing the ChIPs after the 1,6-hexanediol treatment, and show that this leads to a preferential decrease of occupancy at super-enhancers, where the condensates are thought to interact with the genome.

3. Some the authors' terminology is at odds with recent literature, and I suggest they improve the consistency of the language with the literature so as not to confuse readers. For example

3A. "Loop extrusion by CTCF/cohesin" (p2 l18-19, p3 l30). To my knowledge there is no evidence that CTCF mediates loop extrusion. There is indeed mounting evidence that cohesin does. I suggest the authors e.g. simply say "loop extrusion".

3B. "Phase condensate clusters, and these structures" (p9 l13-14, p11, l19, p16 l6). The current consensus view is that clusters of molecules (e.g. Mediator or BRD4) can form condensates through the physical process of phase separation. It is not the condensates that cluster, but the molecules, and the molecule clusters have condensate properties (Banani et al., 2017; Cho et al., 2016; Cho et al., 2018; Shin and Brangwynne, 2017). I suggest the authors e.g. simply say "phase condensates".

3C. "phase separation-mediated assembly of activation complexes" (p4, l22). Mediator is a multiprotein complex that can undergo phase separation, but the assembly of the complex is not known to require phase separation. The authors appear to mean here "phase separation" or "phase separation-mediated assembly of co-activator clusters".

3D. Along the lines above, "BRD4 and coactivators such as MED1" (p4 l29): MED1 is not a coactivator, it is a subunit of the Mediator coactivator (Allen and Taatjes, 2015).

4. I encourage the authors to revise referencing many of their key statements, as some of the referencing seems off.

4A. "Despite the importance of enhancers, very little is known about exactly how they function, although they have been proposed to act mainly as binding platforms for the assembly of protein complexes that can promote gene activation 2,3." There are numerous compressive reviews that synthesize 40 years of literature on enhancer function, which are more appropriate here compared to two primary short reports of the past two years.

4B." Recent work has shown that many enhancer-associated factors, such as Mediator (e.g. MED1) and BRD4, assemble into phase-separated activation complexes, and these interactions appear to be integral to their ability to activate transcription 2,3,34-37". Interestingly enough, none of the cited reports show that phase separation is integral to the ability of Mediator and BRD4 to activate transcription. Indeed most recent perspectives and reviews all argue that the functional role of phase separation in transcription is yet to be demonstrated (McSwiggen et al., 2019).

4C. "Further, several studies have linked the phase separation-mediated assembly of activation complexes at enhancers (particularly SEs) and promoters to the initiation and maintenance of interactions between enhancers and promoters 2,3,35-37." None of those reports included any data supporting the role of condensates in "initiation and maintenance of interactions between enhancers and promoters". Again, condensates are likely to play a role in structuring the genome (Shin et al., 2018), but there is not yet public data linking condensates and enhancer-promoter interactions.

Minor comment

p4 l13 "principal", should be "principle"

References

- Allen, B.L., and Taatjes, D.J. (2015). The Mediator complex: a central integrator of transcription. *Nature reviews Molecular cell biology* 16, 155-166.
- Banani, S.F., Lee, H.O., Hyman, A.A., and Rosen, M.K. (2017). Biomolecular condensates: organizers of cellular biochemistry. *Nature reviews Molecular cell biology*.
- Cho, W.K., Jayanth, N., English, B.P., Inoue, T., Andrews, J.O., Conway, W., Grimm, J.B., Spille, J.H., Lavis, L.D., Lionnet, T., et al. (2016). RNA Polymerase II cluster dynamics predict mRNA output in living cells. *eLife* 5.
- Cho, W.K., Spille, J.H., Hecht, M., Lee, C., Li, C., Grube, V., and Cisse, II (2018). Mediator and

RNA polymerase II clusters associate in transcription-dependent condensates. *Science*.
El Khattabi, L., Zhao, H., Kalchschmidt, J., Young, N., Jung, S., Van Blerkom, P., Kieffer-Kwon, P., Kieffer-Kwon, K.R., Park, S., Wang, X., et al. (2019). A Pliable Mediator Acts as a Functional Rather Than an Architectural Bridge between Promoters and Enhancers. *Cell* 178, 1145-1158 e1120.
McSwiggen, D.T., Mir, M., Darzacq, X., and Tjian, R. (2019). Evaluating phase separation in live cells: diagnosis, caveats, and functional consequences. *Genes & development* 33, 1619-1634.
Shin, Y., and Brangwynne, C.P. (2017). Liquid phase condensation in cell physiology and disease. *Science* 357.
Shin, Y., Chang, Y.C., Lee, D.S.W., Berry, J., Sanders, D.W., Ronceray, P., Wingreen, N.S., Haataja, M., and Brangwynne, C.P. (2018). Liquid Nuclear Condensates Mechanically Sense and Restructure the Genome. *Cell* 175, 1481-1491 e1413.

Reviewer #2 (Remarks to the Author):

The manuscript by Crump et al. describes an in-depth investigation of known/proposed key regulators and mechanisms mediating promoter-enhancer interactions and their direct function in transcriptional gene regulation. Using the leukemia cell line SEM as a model system, the authors first investigate the binding of important factors implicated in promoter-enhancer interactions (BRD2/3/4, MED1, MED12, MED26, CTCF, RAD21) as well as associated histone marks, and also map promoter-enhancer interactions for 62 genes (including known prominent BRD4-dependent genes such as MYC and BCL2) using an elegant high-resolution Capture-C technique. To directly test the role of BRD4 in maintaining these interactions (a widely adopted model in the field), they quantify effects of a well-established BET bromodomain inhibitor (I-BET151) on promoter-enhancer interactions and mRNA output (using an elegant assays for nascent mRNA quantification). Results of these thorough and technically sound analyses are stunning: While I-BET151 induces dramatic changes in mRNA production of analyzed genes that is associated with an eviction of BRD4 and Mediator from chromatin, it has almost no impact on promoter-enhancer interactions. The authors contrast these findings with effects observed after DOT1L inhibition, which indeed affects transcription by disrupting promoter-enhancer interactions and, for the purpose of this study, provides compelling evidence that their assay indeed can detect drug effects on promoter-enhancer interactions with high sensitivity. Crump et al. move on to investigate effects of 1,6-hexanediol, which is commonly used to dissolve phase condensates that have recently been reported to be mechanistically involved in the formation and function of "super-enhancers". Strikingly, while 1,6-hexanediol induced BRD4/Mediator eviction from chromatin and strong direct transcriptional effects, these effects were not accompanied by major changes in promoter-enhancer interactions. In additional studies, the authors demonstrate that CTCF/Cohesin (in contrast to BRD4, Mediator, and phase condensates) are indeed mediating such interactions, which supports a role of the emerging loop-extrusion model as key mechanism forming and maintaining promoter-enhancer interactions.

Overall, the study by Crump et al. describes a technically elegant, sound, and compelling analysis of fundamental mechanisms in chromatin biology and transcriptional gene regulation. Importantly, to test previous models, the authors chose to perform thorough and highly quantitative analyses at selected relevant loci such as MYC and BCL2, which are particularly insightful and distinguish this work from prior studies that mainly relied on genome-wide associations. Based on their in-depth analyses, Crump et al. come to several surprising and highly important conclusions: (1) They convincingly show that stabilization of enhancer-promoter interactions and transcriptional gene regulation are separable events – a finding that on its own deserves publication in a major journal in light of current models. (2) Their findings challenge several models about the mechanistic role of BRD4, BET bromodomain inhibitors, super-enhancers, and phase condensates in chromatin architecture and transcriptional regulation, which in my view have been put forward and widely adopted without rigorous experimental testing. The study by Crump et al. provides exactly such rigorous forward-testing, and results of these experiments challenge some widely adopted models in a very convincing way. The manuscript is very well structured and written and the presented figures support all conclusions in a clear and compelling way. For the first time, I do

not have any major concerns or suggestions that would further improve the quality and impact of this study. I think it should be published as is, and I am convinced that it will have a major impact in the field.

Reviewer #3 (Remarks to the Author):

The study by Crump et al investigates the cofactor requirements for enhancer-promoter contact at several model loci in leukemia cells. The major conclusion of this work is that chemical inhibition of BET bromodomains diminishes transcription yet does not have effects on enhancer-promoter contact. These findings are in accord with recent working that Mediator is dispensable for enhancer-promoter contact (El Khattabi et al, Cell 2019). These findings are timely, but require additional experimental evidence to justify publication.

The major weakness of this study is in relying on compounds, which only causes partial displacement of BET proteins and Mediator from chromatin. As pointed out by the authors, one cannot rule out that these residual levels of these components bound to chromatin are sufficient to retain looping but are insufficient to sustain transcription. With this caveat in mind, the central conclusion of this study is quite weak in its significance to the field. In my view, this work can be significantly strengthened if they can employ a chemical-degradation strategy to cause a more complete eviction of BRD4/Mediator from chromatin, followed by a repeating their chromatin-contact analysis. This could be achieved with PROTACs or using an auxin-degradation allele of BRD4 or MED subunits.

Crump et al Reviewer response

“BET inhibition disrupts transcription but retains enhancer-promoter contact”

We would like to thank all three reviewers for their constructive comments on the paper; they were very helpful and we feel that the paper is now much improved from the original version. We would also like to apologize for the long delay in completing this revision, but, in common with many other institutes across the world, our lab was completely locked down for several months and we are now only partially open. That said, we have been able to generate new data that we hope will help answer most of the concerns raised. In addition, we have rewritten parts of the paper to address other points raised by the reviewers. A specific point by point response is included below.

Reviewers' comments:

Reviewer #1 (Remarks to the Author):

The authors investigated the effect of the inhibition of BET proteins on transcription and enhancer-promoter interactions. They found that in leukemia cells, the ubiquitous co-factors BRD4 and Mediator are enriched at both enhancers and promoters. Treatment of leukemia cells with the BET inhibitor iBET-151 lead to reduced BRD4 binding, rapid decrease of nascent transcription, but a negligible effect on enhancer promoter interactions measured by Capture C. Furthermore, treatment of the cells with 1,6-hexanediol also had negligible effect on enhancer-promoter interactions in contrast to a Dot1l inhibitor.

This paper contributes to a growing body of work that ubiquitous transcriptional co-factors such as BRD4 and Mediator have key functions in controlling transcriptional activity, but have little contribution to interactions between promoters and enhancers [see e.g. (El Khattabi et al., 2019)].

The experiments appear thoroughly performed and analyzed. I have concerns regarding the effect of iBET-151 and the interpretation of the data, the conflict of some of the authors' terminology with the existing literature, and referencing of key statements in the manuscript.

We always strive to reference the literature as accurately as possible and to give scientists their due. We appreciate the reviewer's help in achieving this goal and referencing the literature properly, and we have revised the paper to account for the reviewer's comments.

Specific comments

1. The authors investigated the effect of the inhibition of BET proteins on transcription and enhancer promoter interactions. The key experiment was treating the leukemia cells with the BET inhibitor iBET-151, which lead to a negligible effect on enhancer promoter interactions as assayed by Capture C for about 60 loci. The key concern with this experiment is that the treatment regime of the authors appears to have only a moderate effect on BRD4 binding to the genome, and the majority of the BRD4 ChIP-Seq signal persists after the inhibitor treatment at key loci the authors analyzed (Figure 2f). The results simply do not support the authors' claim that “Loss of BET and Mediator binding is associated with large transcriptional changes at key oncogenic gene targets” (page 7, line 26), although the statement may be ultimately correct [predominantly based on the existing literature, e.g. (El Khattabi et al., 2019)]. The results thus need to be consistently described in the text along the lines of “BET inhibition is associated with large transcriptional changes at key oncogenic gene targets.”

We acknowledge this point, and have addressed it in two different ways. First of all, where appropriate, we have made changes to the text as suggested by the reviewer (e.g. p8 l26-27). In addition, as suggested by reviewer 3, we have conducted additional experiments using the PROTAC molecule AT1 (Gadd et al 2017), which targets BRD4 for proteolytic degradation. We used reference-normalized ChIP-seq to demonstrate that this resulted in a much stronger reduction in the chromatin association of BRD4 and MED1 compared to iBET

treatment, but had no further effect on enhancer-promoter interactions (Supplementary Fig 2e; Fig 2d-g; Fig 3 and Fig 5d). Therefore, we believe we can be more confident in our conclusion that BRD4 is required for transcription of many genes but not maintenance of enhancer-promoter interactions. For the convenience of the reviewer, we have reproduced the key aspects of these new results below.

2. The authors try to instigate that the previously reported ability of Mediator and BRD4 to form phase-separated condensates is linked to the transcriptional effects they observe (e.g. page 9), but this link is not supported by data. For example, the authors write “Consistent

Crump et al Reviewer response

“BET inhibition disrupts transcription but retains enhancer-promoter contact”

with the loss of MED1 foci in cells following BET inhibition^{34,...} (page 9, line 5), and cite the recent paper from the Cisse lab (Cho et al., 2018). First, the Cisse lab used different cells (murine embryonic stem cells), used a different inhibitor (JQ1), used a sophisticated super-resolution-based imaging method to detect condensates in live cells, and did not demonstrate displacement of BRD4 from the genome after their treatment regime. If the authors want to claim any link between condensate formation and their results on transcription, they need to at least perform BRD4 and Mediator immunofluorescence, and show that in the leukemia cells the iBET-151 treatment dissolves BRD4/Mediator foci. Further supporting evidence the authors can quickly collect here is sequencing the ChIPs after the 1,6-hexanediol treatment, and show that this leads to a preferential decrease of occupancy at super-enhancers, where the condensates are thought to interact with the genome.

We take the reviewer’s point that we have not directly linked loss of phase condensate formation with our Capture C experiments. To reflect this, we have carefully rewritten the paper to avoid making any strong claims about condensate formation and enhancer-promoter interactions. For example, we have rewritten the line quoted by the reviewer to “Consistent with the reported interaction of BRD4 and Mediator^{44, 45, 46, 47} (p10 I1-2) with references to several papers that demonstrate the physical association of BRD4 and Mediator. We feel that this is a more relevant point to explain the reduction in MED1 chromatin binding following BET inhibition.

We attempted to replicate the findings of the Cisse lab in our cells, to demonstrate a loss of BRD4/MED1 phase condensates. Unfortunately, the disruption caused by the lab shutdown earlier this year meant that we were unable to generate cell lines expressing fluorescently-labelled BRD4 and MED1, so we could not confirm the loss of BRD4/MED1 foci by live cell imaging.

As an alternative, as suggested by the reviewer, we conducted reference-normalized ChIP-seq for BRD4 and MED1 in untreated and hexanediol-treated SEM cells, and looked at the levels of proteins at super-enhancers. Similar to previous observations (Sabari et al 2018), we found that hexanediol treatment resulted in a stronger reductions in the binding of both proteins to super-enhancers compared to typical enhancers (Fig 4b, below). We note that the effect that we observe with hexanediol treatment is more dramatic than demonstrated by Sabari et al, and suggest that this may be due to our use of reference-normalization of the ChIP-seq, although it is also possible that hexanediol may have a stronger effect in SEM cells compared to mESCs. We also analyzed our nascent RNA-seq data and found a greater reduction in eRNA transcription at super-enhancers compared to typical enhancers (Fig 4b), as well as a greater decrease in expression of genes associated with super-enhancers (Supplementary Fig 4e). Together, these results argue that 1,6-hexanediol acts in our cells as has been previously described in mESCs, dissolving phase condensates at super-enhancers.

Figure 4. a Metaplot of reference-normalized mean BRD4 and MED1 levels at BRD4 peaks in untreated SEM cells (light color) or cells treated with 1.5 % 1,6-hexanediol for 30 min (dark color). b Boxplot showing the log₂ fold-change (logFC) in reference-normalized levels of BRD4 and MED1 and nascent RNA at super-enhancers (SE; olive), or typical enhancers (TE; green) following treatment with 1.5 % 1,6-hexanediol for 30 min. Nascent RNA (eRNA) was measured over 1 kb windows centered on intergenic ATAC-seq peaks overlapping with SEs and TEs. p values indicate the statistical significance of the difference in logFC between SEs and TEs (Wilcoxon rank sum test). Boxplots show median and IQR.

3. Some the authors' terminology is at odds with recent literature, and I suggest they improve the consistency of the language with the literature so as not to confuse readers. For example

3A. “Loop extrusion by CTCF/cohesin” (p2 l18-19, p3 l30). To my knowledge there is no evidence that CTCF mediates loop extrusion. There is indeed mounting evidence that cohesin does. I suggest the authors e.g. simply say “loop extrusion”.

We are sorry for the confusion that our wording caused. For simplicity, we merged the concept that boundaries are CTCF-mediated and loop extrusion is cohesin-mediated, as boundaries are thought to be needed for loop stabilization. However, we can see that this was imprecise so we have rewritten these sections to say “at some sites, CTCF and cohesin” (p2 l19-20) and “chromatin looping mediated by cohesin and bounded by CTCF binding” (p4 l1).

3B. “Phase condensate clusters, and these structures” (p9 l13-14, p11, l19, p16 l6). The current consensus view is that clusters of molecules (e.g. Mediator or BRD4) can form condensates through the physical process of phase separation. It is not the condensates that cluster, but the molecules, and the molecule clusters have condensate properties (Banani et al., 2017; Cho et al., 2016; Cho et al., 2018; Shin and Brangwynne, 2017). I suggest the authors e.g. simply say “phase condensates”.

We have revised these sections to read “phase condensates”, as suggested (new locations p10 l24, p13 l11, p18 l6).

3C. “phase separation-mediated assembly of activation complexes” (p4, l22). Mediator is a multiprotein complex that can undergo phase separation, but the assembly of the complex is not known to require phase separation. The authors appear to mean here “phase separation” or “phase separation-mediated assembly of co-activator clusters”.

We acknowledge the reviewer's point; we had intended to refer to the interaction between BRD4 and Mediator. The phrase has been removed as we rewrote this sentence to address point 4C (below).

3D. Along the lines above, “BRD4 and coactivators such as MED1” (p4 l29): MED1 is not a coactivator, it is a subunit of the Mediator coactivator (Allen and Taatjes, 2015).

We have updated this line to refer to Mediator rather than MED1 (p5 l2-3).

Crump et al Reviewer response

“BET inhibition disrupts transcription but retains enhancer-promoter contact”

4. I encourage the authors to revise referencing many of their key statements, as some of the referencing seems off.

4A. “Despite the importance of enhancers, very little is known about exactly how they function, although they have been proposed to act mainly as binding platforms for the assembly of protein complexes that can promote gene activation 2,3.” There are numerous comprehensive reviews that synthesize 40 years of literature on enhancer function, which are more appropriate here compared to two primary short reports of the past two years.

We have changed these references to cite three recent reviews on enhancers:

Furlong EEM, Levine M. Developmental enhancers and chromosome topology. *Science* 361, 1341-1345 (2018).

Long HK, Prescott SL, Wysocka J. Ever-Changing Landscapes: Transcriptional Enhancers in Development and Evolution. *Cell* 167, 1170-1187 (2016).

Hnisz D, Shrinivas K, Young RA, Chakraborty AK, Sharp PA. A Phase Separation Model for Transcriptional Control. *Cell* 169, 13-23 (2017).

4B.” Recent work has shown that many enhancer-associated factors, such as Mediator (e.g. MED1) and BRD4, assemble into phase-separated activation complexes, and these interactions appear to be integral to their ability to activate transcription 2,3,34-37”. Interestingly enough, none of the cited reports show that phase separation is integral to the ability of Mediator and BRD4 to activate transcription. Indeed most recent perspectives and reviews all argue that the functional role of phase separation in transcription is yet to be demonstrated (McSwiggen et al., 2019).

We have updated this sentence to acknowledge that, whilst a number of papers have suggested a functional relationship between BRD4/MED1 phase separation and transcription, a direct requirement has not been tested:

“Recent work has shown that many enhancer-associated factors, such as Mediator (e.g. MED1) and BRD4, assemble into phase-separated activation complexes, and these interactions are proposed to be integral to their ability to activate transcription^{3, 31, 35, 36, 37, 38, 39}, but a direct requirement for phase condensate formation in transcription has not been established⁴⁰.” (p4 I21-26)

4C. “Further, several studies have linked the phase separation-mediated assembly of activation complexes at enhancers (particularly SEs) and promoters to the initiation and maintenance of interactions between enhancers and promoters 2,3,35-37.” None of those reports included any data supporting the role of condensates in “initiation and maintenance of interactions between enhancers and promoters”. Again, condensates are likely to play a role in structuring the genome (Shin et al., 2018), but there is not yet public data linking condensates and enhancer-promoter interactions.

Our intention with the sentence was to indicate that several papers had implied in their discussion sections that phase separation may be involved in enhancer-promoter contact, although we recognise that our wording was open to the incorrect interpretation that they provided evidence for this. We thank the reviewer for highlighting this, and have rephrased the sentence to it clearer that this function is hypothetical.

“Since these coactivator clusters, which assemble at enhancers (particularly super-enhancers) are also proposed to incorporate promoter-bound RNA polymerase into the condensate^{31, 35, 38, 39}, it is possible that they act as a bridge between these distal DNA

Crump et al Reviewer response

“BET inhibition disrupts transcription but retains enhancer-promoter contact”

elements, and may have a role in initiating and/or maintaining enhancer-promoter interactions^{37, 41, 42}.” (p4 I26-30)

Minor comment

p4 I13 “principal”, should be “principle”

Thank you, we have corrected this.

References for this reviewer

Gadd MS, et al. Structural basis of PROTAC cooperative recognition for selective protein degradation. *Nat Chem Biol* 13, 514-521 (2017).

Sabari BR, et al. Coactivator condensation at super-enhancers links phase separation and gene control. *Science* 361, (2018).

Allen, B.L., and Taatjes, D.J. (2015). The Mediator complex: a central integrator of transcription. *Nature reviews Molecular cell biology* 16, 155-166.

Banani, S.F., Lee, H.O., Hyman, A.A., and Rosen, M.K. (2017). Biomolecular condensates: organizers of cellular biochemistry. *Nature reviews Molecular cell biology*.

Cho, W.K., Jayanth, N., English, B.P., Inoue, T., Andrews, J.O., Conway, W., Grimm, J.B., Spille, J.H., Lavis, L.D., Lionnet, T., et al. (2016). RNA Polymerase II cluster dynamics predict mRNA output in living cells. *eLife* 5.

Cho, W.K., Spille, J.H., Hecht, M., Lee, C., Li, C., Grube, V., and Cisse, II (2018). Mediator and RNA polymerase II clusters associate in transcription-dependent condensates. *Science*.

El Khattabi, L., Zhao, H., Kalchschmidt, J., Young, N., Jung, S., Van Blerkom, P., Kieffer-Kwon, P., Kieffer-Kwon, K.R., Park, S., Wang, X., et al. (2019). A Pliable Mediator Acts as a Functional Rather Than an Architectural Bridge between Promoters and Enhancers. *Cell* 178, 1145-1158 e1120.

McSwiggen, D.T., Mir, M., Darzacq, X., and Tjian, R. (2019). Evaluating phase separation in live cells: diagnosis, caveats, and functional consequences. *Genes & development* 33, 1619-1634.

Shin, Y., and Brangwynne, C.P. (2017). Liquid phase condensation in cell physiology and disease. *Science* 357.

Shin, Y., Chang, Y.C., Lee, D.S.W., Berry, J., Sanders, D.W., Ronceray, P., Wingreen, N.S., Haataja, M., and Brangwynne, C.P. (2018). Liquid Nuclear Condensates Mechanically Sense and Restructure the Genome. *Cell* 175, 1481-1491 e1413.

Reviewer #2 (Remarks to the Author):

The manuscript by Crump et al. describes an in-depth investigation of known/proposed key regulators and mechanisms mediating promoter-enhancer interactions and their direct function in transcriptional gene regulation. Using the leukemia cell line SEM as a model system, the authors first investigate the binding of important factors implicated in promoter-enhancer interactions (BRD2/3/4, MED1, MED12, MED26, CTCF, RAD21) as well as associated histone marks, and also map promoter-enhancer interactions for 62 genes (including known prominent BRD4-dependent genes such as MYC and BCL2) using an elegant high-resolution Capture-C technique. To directly test the role of BRD4 in maintaining these interactions (a widely adopted model in the field), they quantify effects of a well-established BET bromodomain inhibitor (I-BET151) on promoter-enhancer interactions and mRNA output (using an elegant assays for nascent mRNA quantification). Results of these thorough and technically sound analyses are stunning: While I-BET151 induces dramatic changes in mRNA production of analyzed genes that is associated with an eviction of BRD4 and Mediator from chromatin, it has almost no impact on promoter-enhancer interactions. The authors contrast these findings with effects observed after DOT1L inhibition, which indeed affects transcription by disrupting promoter-enhancer interactions and, for the purpose of this study, provides compelling evidence that their assay indeed can detect drug effects on promoter-enhancer interactions with high sensitivity. Crump et al. move on to investigate effects of 1,6-hexanediol, which is commonly used to dissolve phase condensates that have recently been reported to be mechanistically involved in the formation and function of “super-enhancers”. Strikingly, while 1,6-hexanediol induced BRD4/Mediator eviction from chromatin and strong direct transcriptional effects, these effects were not accompanied by major changes in promoter-enhancer interactions. In additional studies, the authors demonstrate that CTCF/Cohesin (in contrast to BRD4, Mediator, and phase condensates) are indeed mediating such interactions, which supports a role of the emerging loop-extrusion model as key mechanism forming and maintaining promoter-enhancer interactions.

Overall, the study by Crump et al. describes a technically elegant, sound, and compelling analysis of fundamental mechanisms in chromatin biology and transcriptional gene regulation. Importantly, to test previous models, the authors chose to perform thorough and highly quantitative analyses at selected relevant loci such as MYC and BCL2, which are particularly insightful and distinguish this work from prior studies that mainly relied on genome-wide associations. Based on their in-depth analyses, Crump et al. come to several surprising and highly important conclusions: (1) They convincingly show that stabilization of enhancer-promoter interactions and transcriptional gene regulation are separable events – a finding that on its own deserves publication in a major journal in light of current models. (2) Their findings challenge several models about the mechanistic role of BRD4, BET bromodomain inhibitors, super-enhancers, and phase condensates in chromatin architecture and transcriptional regulation, which in my view have been put forward and widely adopted without rigorous experimental testing. The study by Crump et al. provides exactly such rigorous forward-testing, and results of these experiments challenge some widely adopted models in a very convincing way. The manuscript is very well structured and written and the presented figures support all conclusions in a clear and compelling way. For the first time, I do not have any major concerns or suggestions that would further improve the quality and impact of this study. I think it should be published as is, and I am convinced that it will have a major impact in the field.

We thank the reviewer for their generous comments. We hope that the additional experiments we've done (BRD4 degradation with AT1 and confirming strong reduction of

Crump et al Reviewer response

“BET inhibition disrupts transcription but retains enhancer-promoter contact”

BRD4/MED1 following AT1 and 1,6-hexanediol treatment) serve to further reinforce the conclusions of the paper.

Reviewer #3 (Remarks to the Author):

The study by Crump et al investigates the cofactor requirements for enhancer-promoter contact at several model loci in leukemia cells. The major conclusion of this work is that chemical inhibition of BET bromodomains diminishes transcription yet does not have effects on enhancer-promoter contact. These findings are in accord with recent working that Mediator is dispensable for enhancer-promoter contact (El Khattabi et al, Cell 2019). These findings are timely, but require additional experimental evidence to justify publication.

The major weakness of this study is in relying on compounds, which only causes partial displacement of BET proteins and Mediator from chromatin. As pointed out by the authors, one cannot rule out that these residual levels of these components bound to chromatin are sufficient to retain looping but are insufficient to sustain transcription. With this caveat in mind, the central conclusion of this study is quite weak in its significance to the field. In my view, this work can be significantly strengthened if they can employ a chemical-degradation strategy to cause a more complete eviction of BRD4/Mediator from chromatin, followed by a repeating their chromatin-contact analysis. This could be achieved with PROTACs or using an auxin-degradation allele of BRD4 or MED subunits.

In order to address the reviewer's concern, we used the PROTAC molecule AT1 (Gadd et al 2017) to target BRD4 for proteolytic degradation. We confirmed a global BRD4 protein depletion by Western blotting (Supplementary Fig 2e), and demonstrated a strong decrease in the chromatin association of both BRD4 and MED1 by reference-normalized ChIP-seq (Fig 2d, f, g). Importantly, BRD4 degradation did not have a stronger effect on enhancer-promoter interactions than IBET, despite the much greater loss of chromatin association (Fig 3b, d, e, f). Whilst it is still possible that the low residual levels of chromatin-associated BRD4 and MED1 are sufficient for chromatin looping (a point we acknowledge in the discussion) we believe that this result further strengthens the conclusions of the paper. We would also like to highlight the fact that both IBET and AT1 are able to disrupt transcription without interfering with enhancer-promoter interactions (Fig 2e), demonstrating that these processes are clearly functionally separate. For the convenience of the reviewer, we have reproduced the key aspects of these new results below.

Gadd MS, et al. Structural basis of PROTAC cooperative recognition for selective protein degradation. Nat Chem Biol 13, 514-521 (2017).

Crump et al Reviewer response
 “BET inhibition disrupts transcription but retains enhancer-promoter contact”

REVIEWERS' COMMENTS

Reviewer #1 (Remarks to the Author):

The revised manuscript appears much improved, and all key concern appear to have been addressed. I recommend acceptance for publication.

Reviewer #2 (Remarks to the Author):

I remain enthusiastic about this elegant and important study and have no further comments.

Reviewer #3 (Remarks to the Author):

the authors have addressed my comments and i now support publication.

REVIEWERS' COMMENTS

We would like to thank the reviewers for their time and efforts and for their kind comments on our paper.

Reviewer #1 (Remarks to the Author):

The revised manuscript appears much improved, and all key concern appear to have been addressed. I recommend acceptance for publication.

Reviewer #2 (Remarks to the Author):

I remain enthusiastic about this elegant and important study and have no further comments.

Reviewer #3 (Remarks to the Author):

the authors have addressed my comments and i now support publication.